# Annotation-efficient deep learning for automatic medical image segmentation

Shanshan Wang 1,2,3,11✉, Cheng Li 1,11✉, Rongpin Wang4,11, Zaiyi Liu 5, Meiyun Wang 6, Hongna Tan6, Yaping Wu6, Xinfeng Liu4, Hui Sun1, Rui Yang7, Xin Liu1, Jie Chen2,8, Huihui Zhou9, Ismail Ben Ayed10 & Hairong Zheng 1✉

Automatic medical image segmentation plays a critical role in scientific research and medical care. Existing high-performance deep learning methods typically rely on large training datasets with high-quality manual annotations, which are difficult to obtain in many clinical applications. Here, we introduce Annotation-effIcient Deep lEarning (AIDE), an open-source framework to handle imperfect training datasets. Methodological analyses and empirical evaluations are conducted, and we demonstrate that AIDE surpasses conventional fully-supervised models by presenting better performance on open datasets possessing scarce or noisy annotations. We further test AIDE in a real-life case study for breast tumor segmentation. Three datasets containing 11,852 breast images from three medical centers are employed, and AIDE, utilizing 10% training annotations, consistently produces segmentation maps comparable to those generated by fully-supervised counterparts or provided by independent radiologists. The 10-fold enhanced efficiency in utilizing expert labels has the potential to promote a wide range of biomedical applications.

[1] Paul C. Lauterbur Research Center for Biomedical Imaging, Shenzhen Institutes of Advanced Technology, Chinese Academy of Sciences, Shenzhen, Guangdong, China. [2] Peng Cheng Laboratory, Shenzhen, Guangdong, China. [3] Pazhou Laboratory, Guangzhou, Guangdong, China. [4] Department of Medical Imaging, Guizhou Provincial People's Hospital, Guiyang, Guizhou, China. [5] Department of Medical Imaging, Guangdong General Hospital, Guangdong Academy of Medical Sciences, Guangzhou, Guangdong, China. [6] Department of Medical Imaging, Henan Provincial People's Hospital & the People's Hospital of Zhengzhou University, Zhengzhou, Henan, China. [7] Department of Urology, Renmin Hospital of Wuhan University, Wuhan, Hubei, China. [8] School of Electronic and Computer Engineering, Shenzhen Graduate School, Peking University, Shenzhen, Guangdong, China. [9] Brain Cognition and Brain Disease Institute, Shenzhen Institutes of Advanced Technology, Chinese Academy of Sciences, Shenzhen, Guangdong, China. [10] ETS Montreal, Montreal, Canada. [11] These authors contributed equally: Shanshan Wang, Cheng Li, Rongpin Wang. ✉email: ss.wang@siat.ac.cn; cheng.li6@siat.ac.cn; hr.zheng@siat.ac.cn

Medical imaging contributes significantly to progress in scientific discoveries and medicine[1]. Semantic segmentation partitions raw image data into structured and meaningful regions and thus enables further image analysis and quantification, which are critical for various applications, including anatomy research, disease diagnosis, treatment planning, and prognosis monitoring[2–5]. With the global expansion of medical imaging and the advancement of imaging techniques, the volume of acquired medical image data is increasing at a pace much faster than available human experts can interpret. Thus, automated segmentation algorithms are needed to assist physicians in realizing accurate and timely imaging-based diagnosis[6,7].

In the past decade, deep learning has made considerable progress in automatic medical image segmentation by demonstrating promising performance in various breakthrough studies[8–13]. Nevertheless, the applicability of deep-learning methods to clinical practice is limited because of the heavy reliance on training data, especially training annotations[14,15]. Large curated datasets are necessary, but annotating medical images is a time-consuming, labor-intensive, and expensive process. Depending on the complexity of the regions of interest to segment and the local anatomical structures, minutes to hours may be required to annotate a single image. Furthermore, label noise is inevitable in real-world applications of deep-learning models[16]. Such noise can result from systematic errors of the annotator, as well as inter-annotator variation. More than three domain experts are typically needed to generate trustworthy annotations[17]. Any biases in the data can be transferred to the outcomes of the learned models[18]. Consequently, the lack of large and high-quality labeled datasets has been identified as the primary limitation of the application of supervised deep learning for medical imaging tasks[19–21]. Learning with imperfect datasets having limited annotations (semi-supervised learning, SSL), lacking target domain annotations (unsupervised domain adaptation, UDA), or containing noisy annotations (noisy label learning, NLL) are three of the most frequently encountered challenges in clinical applications[22].

Co-training is one of the most prevalent methods for SSL[23,24] that works by training two classifiers for two complementary views using the labeled data, generating pseudo-labels for unlabeled data by enforcing agreement between the classifier predictions, and combining the labeled and pseudo-labeled data for further training. Co-training has been employed mainly to semi-supervised classification tasks. Only recently has co-training been extended to semi-supervised image segmentation[25] and UDA of segmentation models[26]. Despite the achieved inspiring performance, the direct employment of co-training methods to NLL is problematic, as they do not possess the ability to distinguish between accurate and noisy labels[26,27]. Co-teaching was developed based on co-training to specifically address the challenge of NLL by dropping suspected highly noisy samples during network optimization[27,28]. Nevertheless, data filtering by dropping samples is an inefficient approach that might lead to model learning from a pseudo-realistic data distribution; thus, co-teaching methods are more applicable to natural image classification tasks when sufficient large datasets are available to cover the full range of different situations, even after data dropping.

In this study, we develop an annotation-efficient deep-learning framework for medical image segmentation, which we call AIDE, to handle different types of imperfect datasets. AIDE is designed to address all three challenges of SSL, UDA, and NLL. With AIDE, SSL and UDA are transformed into NLL by generating low-quality noisy labels for the unlabeled training data utilizing models trained either on the limited annotated data (SSL) or on the annotated source domain data (UDA). A cross-model self-correction method is proposed to achieve annotation-efficient network learning. Specifically, cross-model co-optimization learning is realized by training two networks in parallel and conducting cross-model information exchange. With the exchanged information, self-label filtering and correction of inexpensive noisy labels proceed with cascaded local and global steps progressively in an elaborately designed schedule according to an observed small loss criterion. The framework is flexible regarding the deep neural network (DNN) models to be utilized.

Methodological analyses are performed to evaluate the effectiveness of AIDE for handling imperfect training datasets. In order to fairly evaluate the method without severe ground-truth label biases, we conduct extensive experiments on a variety of public datasets which have widely accepted data and labels. Specifically, extensive experiments on open datasets with limited, no target domain, and noisy annotations are performed. Better results are achieved with AIDE compared to the respective fully supervised baselines optimized with the available noisy annotations or model-generated low-quality labels. Furthermore, to test the applicability of our method in real-world applications, three datasets for breast tumor segmentation (11,852 image samples of 872 patients) collected from three medical centers are experimented with. The annotations of these data are carefully curated by three experienced radiologists with more than 10 years of experience in breast MR image interpretation. Comparable segmentation results to those obtained by fully supervised models with access to 100% training data annotations and those provided by independent radiologists are achieved with AIDE by utilizing only 10% of the annotations. Our results indicate that DNNs are capable of exploring the image contents of large datasets under proper guidance without the necessity of high-quality annotations. We believe that our proposed framework has the potential to improve the medical image diagnosis workflow in an efficient manner and at a low cost.

## Results

**AIDE: a deep-learning framework to handle imperfect training datasets.** AIDE is a deep-learning framework that achieves accurate image segmentation with imperfect training datasets. A cross-model self-label correction mechanism is proposed to efficiently exploit inexpensive noisy training data labels, which are either generated by pretrained low-performance deep-learning models or provided by individual annotators without quality controls.

An overview of AIDE and example images of the datasets we utilized are depicted in Fig. 1. AIDE is proposed to address three challenging tasks caused by imperfect training datasets (Fig. 1a). The first is SSL with limited annotated training data. Proper utilization of the relatively abundant unlabeled data is very important in this case. The second is UDA, where large discrepancies may exist between the target and source domains. The third is NLL that considers the variations of annotations provided by different observers. By means of task standardization by generating low-quality noisy labels for SSL and UDA (for SSL, models are pretrained with the available limited annotated training data, and low-quality labels are generated for the remaining unlabeled training data; for UDA, models are pretrained with the source domain labeled training data, and low-quality labels are generated for the target domain unlabeled training data), all three challenging tasks are addressed in one framework that targets model optimization by datasets containing problematic labels (Fig. 1a).

For AIDE, two networks are trained in parallel to conduct cross-model co-optimization (Fig. 1b). In each iteration, those samples in an input batch that are suspected to have noisy labels are filtered out and are subjected to data augmentation (random rotation and flipping), and corresponding pseudo-labels are generated by distilling the predictions of the augmented inputs (local label filtering in Fig. 1b). These pseudo-labels, together with

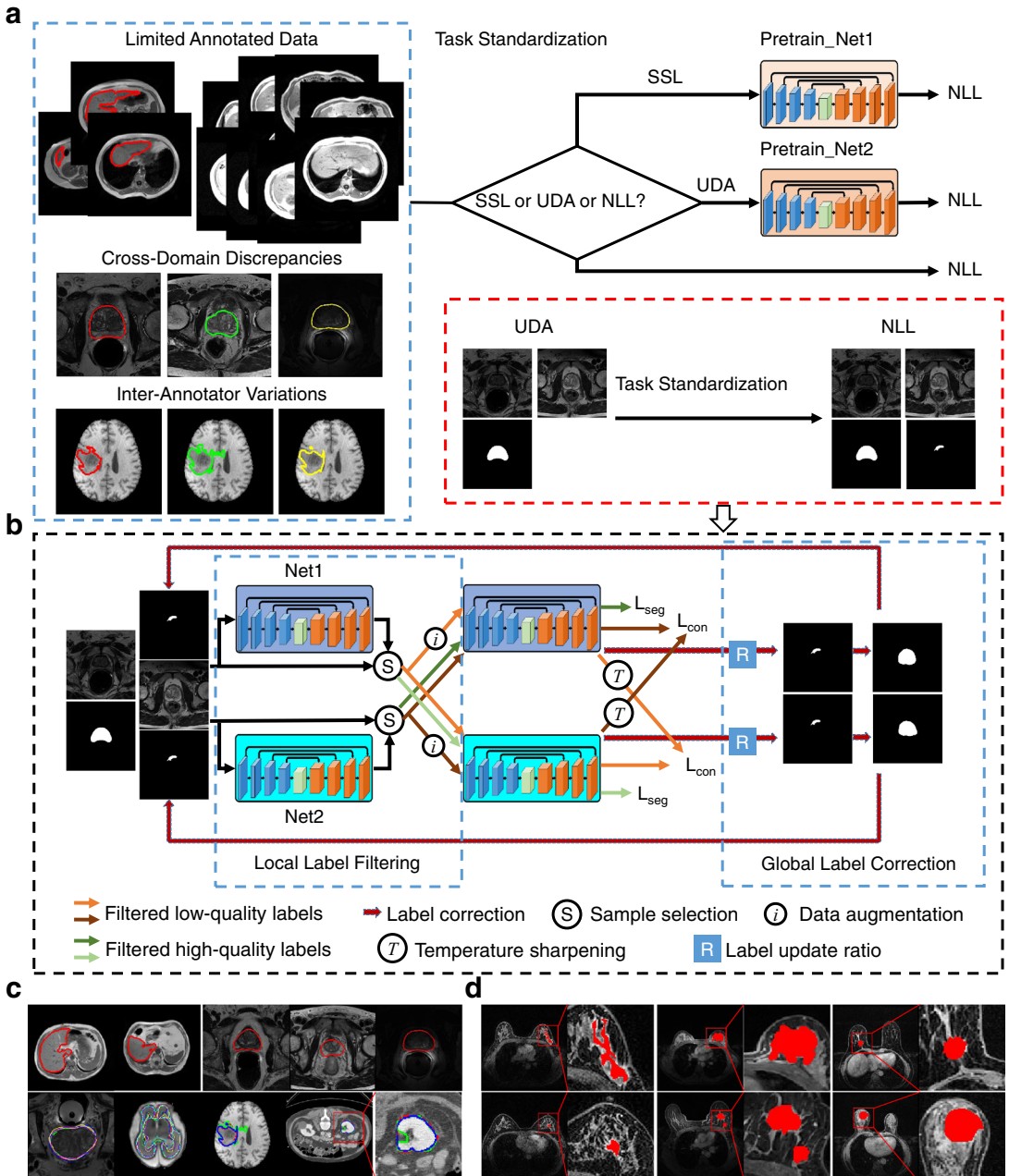

**Fig. 1 Overview of AIDE. a** The three challenges (semi-supervised learning (SSL), unsupervised domain adaptation (UDA), and noisy label learning (NLL)) that AIDE addresses and the proposed task standardization method. **b** The overall framework of AIDE, which comprises three major elements: local label filtering, global label correction, and cross-model co-optimization. **c** Example images of the open datasets. The top left two images are from the CHAOS dataset. The top right three images are from the three domains of prostate datasets. The bottom images are from the QUBIQ datasets, where the first four correspond to the four subtasks and the last one is an enlarged view of the fourth image. Color lines indicate the target regions. **d** Example images of the breast datasets. From left to right, the three columns correspond to the images collected from the three medical centers. Red color regions show the breast tumors.

the high-quality labels, are utilized to train the network. After each epoch, the labels of the whole training set are analyzed, and those that have low similarities (smaller Dice scores) with the network's predictions are updated if an elaborately designed updating criterion is met (global label correction in Fig. 1b). With AIDE, the networks are forced to concentrate on the image contents instead of extracting image features guided by only the annotations. The exact segmentation network we employ has a classical encoder−decoder structure with multiple streams to extract image features from different modalities[29,30], should multimodal inputs be available.

Multiple evaluation metrics are reported to characterize the segmentation performance, namely, the Dice score (Dice similarity coefficient, DSC), relative area/volume difference (RAVD), average symmetric surface distance (ASSD), and maximum symmetric surface distance (MSSD). Higher DSC values and lower RAVD, ASSD, and MSSD indicate more accurate segmentation results.

**Enhancement of AIDE compared to conventional fully supervised learning**. The effectiveness of AIDE in handling low-quality annotated data can be attributed to the following advantages.

First, the local label-filtering step in each iteration enforces a constraint on suspected low-quality annotated samples to generate predictions that are consistent with those of the augmented inputs, which is a form of data distillation that enforces a transformation-consistent constraint on the model predictions[31]. This design can prevent the negative effects of low-quality annotations and exploit as much useful information as possible from these inputs instead of dropping them. Compared to conventional fully supervised learning, which trains the network only by minimizing the discrepancies between the predictions and the ground-truth labels, the consistency constraint of AIDE can force the network to learn latent features from inputs that are transformation invariant; in this way, the network concentrates more on the image contents instead of focusing on only the regression of ground-truth labels. Second, the global label-correction step after each epoch progressively shifts the networks towards predicting consistent results at different time points. In other words, the networks are expected to have small variations, and those samples that lead to large variations are considered low-quality annotated samples since low-quality labels, to a certain extent, contradict the information provided in the images. These suspected low-quality annotated samples should be corrected. Direct label correction to replace the old labels, which are predicted by the framework in previous steps, aims to find useful information from the framework's own outputs; thus, it offers the framework a self-evolving capability. Since in most cases, medical images of the same region appear roughly similar among different patients, we believe that this evolving capability of updating a portion of suspected low-quality annotations is reasonable and applicable. The remaining non-updated samples can guarantee the stable training of the network. Third, the proposed cross-model co-optimization learning can prevent error propagation and accumulation in one network. By building two networks and letting them exchange information, we reduce the risk of network overfitting to its own predicted pseudo-labels.

**Semi-supervised learning with severely limited training samples**. The CHAOS dataset, which is built for liver segmentation, is introduced to investigate the effectiveness of AIDE for SSL (Fig. 1c). Different sizes of high-quality annotated image samples are utilized to train the networks. As expected, a greater number of training samples results in improved segmentation results (Supplementary Fig. 1 and Supplementary Table 1). Compared to that of networks trained with all ten labeled cases (331 image samples, DSC: 87.9%, RAVD: 10.4%, ASSD: 4.65 mm, and MSSD:

65.1 mm), the segmentation performance of networks trained with only one labeled case (30 image samples) is significantly worse (DSC: 70.1%, RAVD: 16.1%, ASSD: 16.1 mm, and MSSD: 176.3 mm). Thus, when training deep-learning models utilizing only the labeled samples in a fully supervised optimization manner, large datasets are required to obtain satisfactory results.

To enlarge the training dataset, low-quality noisy labels are generated by models trained with the minimal annotated training data (1 case with 30 image samples). Then, networks are trained with the enlarged dataset. In this work, low-quality labels or noisy labels are counted in an image-based manner. Different levels of noisy labels are experimented with (Supplementary Table 1). For the fully supervised learning baseline, noisy labels already affect their performance when the noise level is over 20% ($P = 0.0133$ from a two-sided paired $t$ test between the DSCs of Exp. 6 and Exp. 7 in Supplementary Table 1). Under the experimental settings of over 90% noisy labels (S05_30_301_F_NP and S09_30_954_F_NP in Table 1), the segmentation performance of the baseline method deteriorates dramatically. Our proposed AIDE is effective when limited annotations are available and low-quality noisy labels are utilized (Table 1). It achieves much better results (S12_30_954_A_YP in Table 1) compared to those achieved by the baseline method (S10_30_954_F_YP in Table 1) and those obtained by existing methods (pseudo-label[32] and co-teaching[27] in Table 1) when the same quantity of labeled training data is utilized (the pseudo-label method is different from our setting of S10_30_954_F_YP in that the low-quality labels are updated during model optimization for the pseudo-label method, whereas for S10_30_954_F_YP, the low-quality labels remain unchanged once generated). Promising performance is achieved by our method with high noise levels such as the 97% noise level experimented with (S12_30_954_A_YP in Table 1). Particularly, with 30 labeled and 954 unlabeled training image samples (noise level of 97%), our method generates comparable results (DSC: 86.9%, RAVD: 10.0%, ASSD: 4.17 mm, and MSSD: 44.6 mm) to those of models trained with 331 high-quality annotated samples (DSC: 88.5%, RAVD: 10.8%, ASSD: 3.64 mm, and MSSD: 43.8 mm) (no significant difference between the DSCs, with $P = 0.0791$ from a two-sided paired $t$ test).

Different numbers of labeled training image samples have been experimented with, and the results confirm that our method is always effective (Exp. 14 to Exp. 19 in Supplementary Table 1). In addition, as the number of unlabeled training samples increases, the performance of our method (compared that of S12_30_954_A_YP with S08_30_301_A_YP in Table 1) continues to improve. This

**Table 1 Segmentation results of networks under different SSL settings.**

| Settings | Train HQA | Train LQA | AIDE | PP | DSC (%) | RAVD (%) | ASSD (mm) | MSSD (mm) |
|---|---|---|---|---|---|---|---|---|
| S01_30_0_F_NP | 30 | 0 | No | No | 70.1 | 42.0 | 16.1 | 176.3 |
| S02_30_0_F_YP | 30 | 0 | No | Yes | 75.6 | 22.0 | 7.68 | 54.8 |
| S03_331_0_F_NP | 331 | 0 | No | No | 87.9 | 10.4 | 4.65 | 65.1 |
| S04_331_0_F_YP | 331 | 0 | No | Yes | 88.5 | 10.8 | 3.64 | 43.8 |
| S05_30_301_F_NP | 30 | 301 | No | No | 78.4 | 19.0 | 7.92 | 95.3 |
| S06_30_301_F_YP | 30 | 301 | No | Yes | 79.7 | 21.1 | 6.10 | 53.4 |
| S07_30_301_A_NP | 30 | 301 | Yes | No | 79.8 | 18.5 | 10.8 | 116.3 |
| S08_30_301_A_YP | 30 | 301 | Yes | Yes | 82.9 | 16.9 | 5.43 | 49.8 |
| S09_30_954_F_NP | 30 | 954 | No | No | 79.5 | 19.7 | 8.43 | 104.2 |
| S10_30_954_F_YP | 30 | 954 | No | Yes | 80.2 | 19.1 | 6.47 | 53.8 |
| S11_30_954_A_NP | 30 | 954 | Yes | No | 86.1 | 10.2 | 5.49 | 75.8 |
| S12_30_954_A_YP | 30 | 954 | Yes | Yes | 86.9 | 10.0 | 4.17 | 44.6 |
| Pseudo-label[32] | 30 | 954 | No | Yes | 81.4 | 19.7 | 5.99 | 52.9 |
| Co-teaching[27] | 30 | 954 | No | Yes | 82.4 | 15.8 | 5.50 | 49.2 |

HQA and LQA indicate high-quality and low-quality annotations. LQAs are generated by the model trained using the data provided with HQAs. PP refers to post-processing, which is the process of keeping the largest connected components. Context is included in the notation of the experimental setting. For the setting S05_30_301_F_NP, 30 refers to 30 training samples with HQAs, 301 means 301 training samples with LQAs, F indicates training with the conventional fully supervised learning approach, and NP means no post-processing.

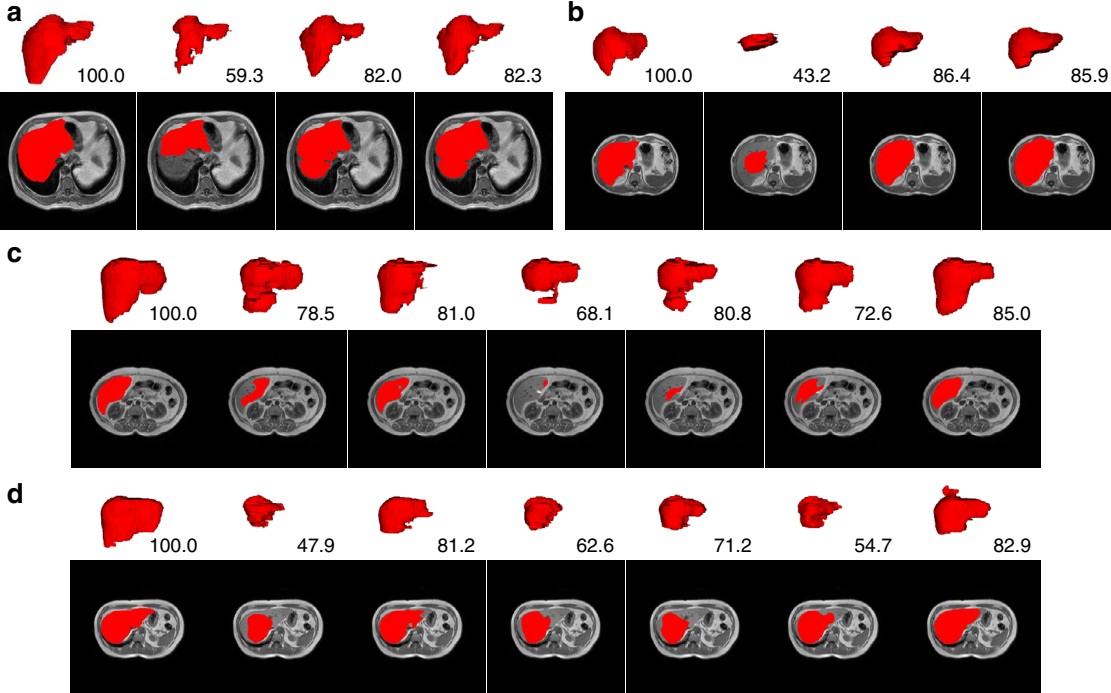

**Fig. 2 Visualizations of label correction and segmentation results for SSL. a, b** Example results of training data label correction. The red regions in the four images from left to right correspond to the high-quality label, the low-quality label utilized to train the model, and the self-corrected labels of the two networks, respectively. **c, d** Example segmentation results. The first columns are the corresponding high-quality labels. The second to the last columns are the results achieved under settings S02, S04, S06, S08, S10, and S12 in Table 1. The numbers are the DSC values (%). In each subfigure, the first row (white background) shows the segmentation results in 3D rendering, and the second row (black background) gives the results of a single selected slice in 2D.

**Table 2 Segmentation results of networks trained and tested with prostate datasets of different domains.**

| Training dataset | Testing dataset | DSC (%) | RAVD (%) | ASSD (mm) | MSSD (mm) |
|---|---|---|---|---|---|
| Domain 1 | Domain 1 | 88.9 | 11.1 | 1.49 | 8.27 |
| Domain 1 | Domain 2 | 45.8 | 65.6 | 5.42 | 17.8 |
| Domain 1 | Domain 3 | 40.0 | 67.8 | 9.24 | 24.3 |
| Domain 2 | Domain 1 | 69.8 | 35.5 | 3.43 | 12.9 |
| Domain 2 | Domain 2 | 87.3 | 16.1 | 1.36 | 7.89 |
| Domain 2 | Domain 3 | 55.1 | 53.0 | 8.32 | 21.3 |
| Domain 3 | Domain 1 | 80.9 | 8.61 | 2.43 | 14.9 |
| Domain 3 | Domain 2 | 86.1 | 12.3 | 1.58 | 8.27 |
| Domain 3 | Domain 3 | 86.7 | 8.77 | 1.55 | 9.20 |

property is especially important for clinical applications, as unlabeled data are much easier to collect. Visualizations of the self-corrected labels and model outputs are shown in Fig. 2 (more results can be found in Supplementary Fig. 2). Overall, the labels after correction are closer to the high-quality annotations (Fig. 2a, b), and our AIDE generates more accurate segmentations than do the corresponding baseline networks (Fig. 2c, d).

Results generated by AIDE, which is trained with ten training cases (one labeled case and nine unlabeled cases), are submitted for evaluation online, and an average DSC of 83.1% is achieved on the test data. This DSC value is within the range of the values achieved by fully supervised methods trained on 20 labeled cases ([63%, 95%] as reported in the summary paper of the CHAOS challenge[33]). Considering the small labeled image data utilized, our method achieves a reasonably good performance compared to these fully supervised ones.

**Unsupervised domain adaptation with large domain discrepancies.** Three domains (domains 1, 2, and 3) of prostate datasets with different image acquisition protocols are utilized to inspect the framework performance when no target domain annotations are provided during model training. Domain 3 is a combined dataset with different image acquisition parameters. When training with a single domain dataset, the networks become biased to the domain properties, and the performance on data from other domains is compromised. Table 2 presents the results of networks tested on the same domain or different domains from the training set. Obvious reductions in segmentation accuracy are observed when cross-domain testing (especially for models optimized in domain 1 or domain 2) is performed. Meanwhile, when labeled training data from a different domain are included during model training, the model performance improves slightly (Supplementary Table 3).

Similar to the strategies adopted for SSL, low-quality noisy labels for the target domain training data are generated by the models trained with the source domain labeled data, and model training from the scratch with the combined dataset is conducted to facilitate the model adaptation to new domains. Figure 3 presents the prostate segmentation results of models optimized

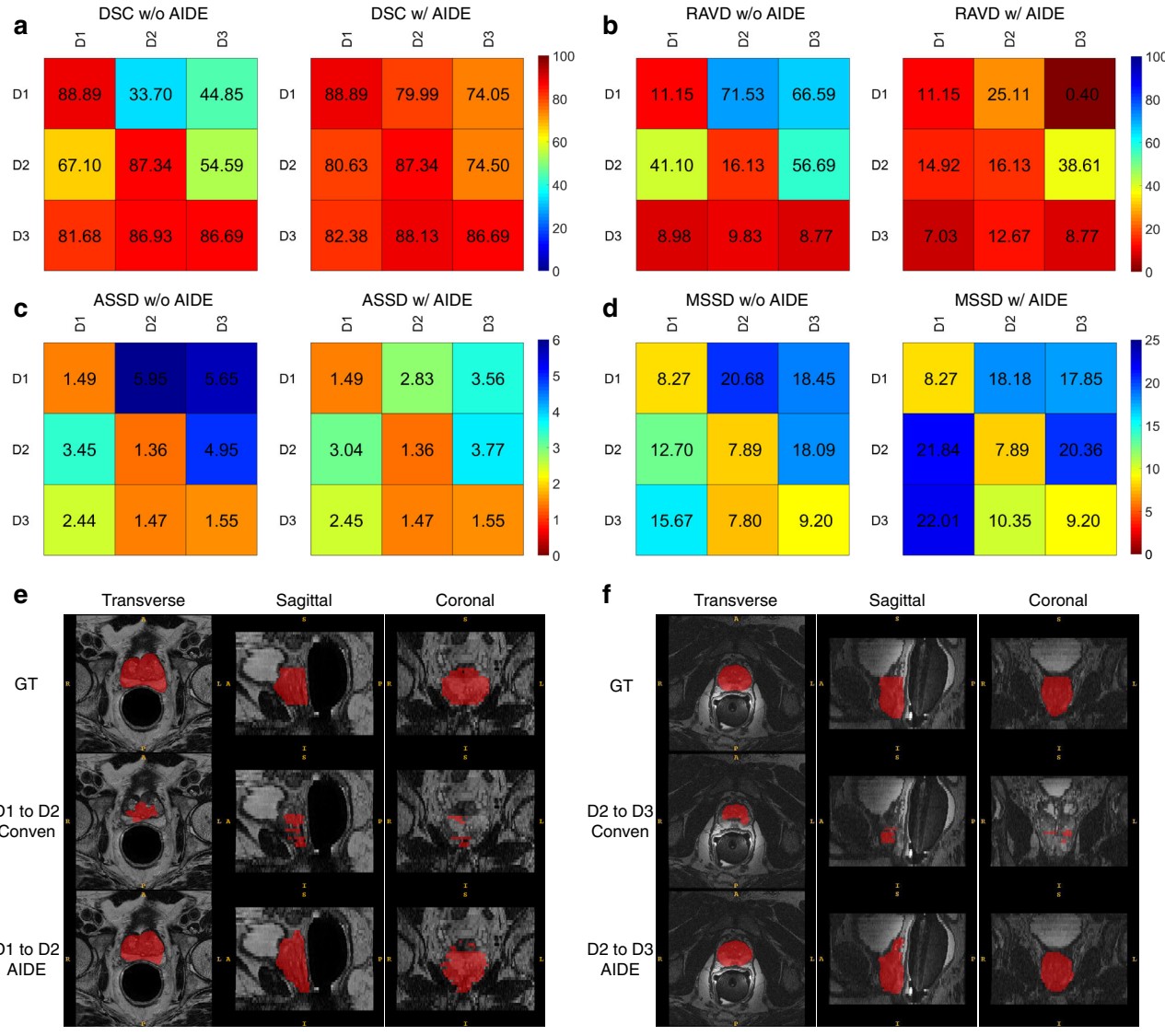

**Fig. 3 Results of prostate segmentation for UDA. a–d**, The four evaluation metrics, DSC (%), RAVD (%), ASSD (mm), and MSSD (mm). In each subfigure, the left and right mappings indicate that the networks are trained without and with the proposed AIDE. D1, D2, and D3 refer to domains 1, 2, and 3. In each mapping, the vertical axis indicates the dataset utilized to train the models and the horizontal axis indicates the dataset utilized to test the models. **e–f** Example segmentation results when transferring models between domains. GT stands for ground truth, referring to the high-quality annotations. Conven is conventional, indicating that the results are generated by a fully supervised optimization method utilizing the combined dataset (high-quality labels for the source domain training data and model-generated low-quality labels for the target domain training data).

without target domain high-quality training annotations, with or without the proposed AIDE framework (more results can be found in Supplementary Figs. 3–5 and Supplementary Table 3). AIDE successfully improves the segmentation performance, as indicated by the large increase in DSC (Fig. 3a) and the notable decrease in RAVD (Fig. 3b). As a special case, when transferring models from domain 1 to domain 2, direct model training with the combined dataset utilizing high-quality annotations for domain 1 and model-generated low-quality labels for domain 2 produces worse results (DSC: 33.7%, RAVD: 71.5%, ASSD: 5.94 mm, and MSSD: 20.7 mm) than those achieved by direct testing of domain 1 optimized models on domain 2 data (DSC: 45.8%, RAVD: 65.6%, ASSD: 5.42 mm, and MSSD: 17.8 mm). On the other hand, AIDE increases DSC by more than 30% (from 45.8 to 80.0%). However, the distance metrics (ASSD in Fig. 3c and MSSD in Fig. 3d) are not largely improved. In addition, even with AIDE, the performance on domain 3 is worse than that on domain 1 and domain 2. Since domain 3 is a combination of data

from different sources, we speculate that adapting models to a combined diverse dataset is more difficult. Nevertheless, the performance is acceptable, and the DSC values fall in the range of reported results (from 0.71 to 0.90)[34]. Besides, better performance is achieved by AIDE than that of the two existing methods, pseudo-label[32] and co-teaching[27] (Supplementary Table 4). Overall, our experiments validate that the proposed framework achieves promising results for UDA.

**Noisy label learning with annotations provided by different annotators.** Four subtasks, including prostate segmentation, brain growth segmentation, brain tumor segmentation, and kidney segmentation, are investigated using the QUBIQ datasets when multiple annotations are provided. Variations exist between annotations provided by different experts, especially for small target objects (Fig. 1c). We treat each set of annotations provided by respective annotators as noisy labels. Overall, testing with

different thresholds shows that AIDE can generate segmentation result distributions that are consistent with the annotators (Tables 3–5 and Supplementary Figs. 6 and 7). Although the improvements on relatively large region segmentation tasks (Task1_Prostate Segmentation in Table 3 and Task2_Brain Growth Segmentation in Table 4) are minor, AIDE achieves better performance on the more challenging small object segmentation tasks (Task3_Brain Tumor Segmentation and Task4_Kidney Segmentation in Table 5). Online server evaluation gives average DSCs of 93.2%, 86.6%, 93.7%, and 89.8% on the four tasks for our method optimized with only a single set of annotations from one annotator for each task. These results are comparable to those achieved by methods utilizing the annotations provided by multiple annotators.

DNNs possess a high capacity to fit training data, even in the presence of noise. Two or even three annotators may not be sufficient to cover the full range of observer variability[17]. To force the model to learn from the real data distributions instead of learning possible biases introduced by annotators, for conventional fully supervised deep-learning models, trustworthy labels obtained by effectively combining the annotations provided by different observers are needed. With our proposed method, only noisy labels are utilized during the whole training process, and the model can self-adjust and update the training labels via the proposed local label-filtering and global label-correction steps to correct possible observer biases. More satisfactory segmentation results can then be generated. Therefore, compared to the fully supervised learning approach, AIDE may relax the demand for multiple annotators.

**Real-life case study for breast tumor segmentation**. Further evaluations on three breast tumor segmentation datasets (GGH dataset from Guangdong General Hospital, GPPH dataset from Guizhou Provincial People's Hospital, and HPPH dataset from Henan Provincial People's Hospital) are conducted to investigate the feasibility of the proposed framework for processing raw clinical data. Dynamic contrast-enhanced MR (DCE-MR) images are acquired. The experimental results validate the effectiveness of AIDE in analyzing clinical samples with decreased manual efforts (Fig. 4). Under the same experimental settings, AIDE generates much better results than the respective baselines, increasing the DSC values by more than 7% (7.7% absolute increase for GGH dataset between LQA200 and LQA200_Ours in Fig. 4a, 18.2% absolute increase for GPPH dataset between LQA100 and LQA100_Ours in Fig. 4b, and 10.0% absolute increase for HPPH dataset between LAQ272 and LQA272_Ours in Fig. 4c). In addition, despite the small number of training annotations being utilized (10% of those utilized by fully supervised method for GGH and GPPH, and 9.2% for HPPH), AIDE can always achieve segmentation performance similar to that of the corresponding fully supervised models. Specifically, for the GGH dataset, AIDE achieves an average DSC of $0.690 \pm 0.251$, whereas the fully supervised model achieves $0.722 \pm 0.208$ ($P = 0.0608$) (Fig. 4a). For GPPH, AIDE obtains $0.654 \pm 0.221$, and the fully supervised model obtains $0.678 \pm 0.260$ ($P = 0.2927$) (Fig. 4b). For HPPH, AIDE obtains $0.731 \pm 0.196$, and the fully supervised model obtains $0.738 \pm 0.227$ ($P = 0.6545$) (Fig. 4c). The visual results confirm the effectiveness of AIDE in generating segmentation contours that are similar to the ground-truth labels (Fig. 4d–f). We further observe that AIDE performs better with larger datasets (comparing Fig. 4c to Fig. 4b), which can be very useful in real clinical applications considering the large quantities of unannotated images accumulated every day.

Additional radiologists were employed to segment the breast tumors in the central slices of the three test sets (Fig. 4d–f and Supplementary Figs. 8–10). For the GGH dataset, the manual annotations achieve an average DSC of $0.621 \pm 0.155$, which is worse than that of AIDE ($0.690 \pm 0.251$), with $P = 0.0098$. For GPPH, the manual annotations obtain $0.861 \pm 0.086$ and AIDE obtains $0.846 \pm 0.118$. No significant difference is found ($P = 0.3317$). For HPPH, the manual annotations obtain $0.735 \pm 0.225$ and AIDE obtains $0.761 \pm 0.234$ with no significant difference observed ($P = 0.3079$). Overall, our proposed AIDE can generate comparable or even better segmentation results when compared with those of radiologists.

Breast tumor segmentation in DCE-MR images is a challenging task due to the severe class imbalance issue (very small tumor regions compared to the whole breast images) and the high confounding background signals (organs in the chest and dense glandular tissues). As a result, the DSC values obtained for breast tumor segmentation are not as high as those for other tasks (such as prostate segmentation). Nevertheless, our method is not intended to replace radiologists in the disease diagnosis or treatment planning workflow but serves as an automated computer-aided system. Recently, Zhang et al.[35] achieved a mean DSC of 72% for breast tumor segmentation in DCE-MR images with a hierarchical convolutional neural network framework, and Qiao et al.[36] obtained a value of 78% utilizing 3D U-Net and single-phase DCE-MR images. Our results (73.1% on the HPPH dataset) are comparable to these literature-reported values achieved by fully supervised learning with hundreds of patient cases. Considering the small number of training annotations utilized (25 cases for the HPPH dataset), our proposed method can be a valuable tool in clinical practice to assist radiologists in achieving fast and reliable breast tumor segmentation.

## Discussion

Image segmentation plays an important role in medical imaging applications. In recent studies, DNNs have been widely employed to automate and accelerate the segmentation process[37–40]. Despite the reported successful applications, the performance of DNNs depends heavily on the quality of the utilized training datasets. Frameworks that can optimize DNNs without strict reliance on large and high-quality annotated training datasets can significantly narrow the gap between research and clinical

---

**Table 3 Segmentation performance (DSC: %) of Task1_Prostate Segmentation on the QUBIQ datasets.**

| Method | Anno1 | Anno2 | Anno3 | Anno4 | Anno5 | Anno6 |
|---|---|---|---|---|---|---|
| Conventional | 89.2 | 87.9 | 84.3 | 89.3 | 86.4 | 90.0 |
| AIDE | 91.4 | 89.2 | 85.2 | 90.4 | 88.1 | 90.3 |

Conventional method refers to the fully supervised method utilizing labels of the respective annotations (e.g., Anno1). Anno1 to Anno7 indicate the annotations utilized to train the model (e.g., Anno1 means annotations from annotator 1 are used).

---

**Table 4 Segmentation performance (DSC: %) of Task2_Brain Growth Segmentation on the QUBIQ datasets.**

| Method | Anno1 | Anno2 | Anno3 | Anno4 | Anno5 | Anno6 | Anno7 |
|---|---|---|---|---|---|---|---|
| Conventional | 70.7 | 73.2 | 73.8 | 73.2 | 71.6 | 72.1 | 71.8 |
| AIDE | 71.1 | 75.0 | 74.8 | 73.5 | 71.6 | 72.7 | 72.0 |

**Table 5 Segmentation performance (DSC: %) of Task3_Brain Tumor Segmentation and Task4_Kidney Segmentation on the QUBIQ datasets.**

| Method | Anno1 | Anno2 | Anno3 |
|---|---|---|---|
| Task3_Brain Tumor Segmentation | | | |
| Conventional | 83.9 | 83.1 | 84.2 |
| AIDE | 92.6 | 91.8 | 92.7 |
| Task4_Kidney Segmentation | | | |
| Conventional | 69.8 | 68.9 | 70.7 |
| AIDE | 85.4 | 83.8 | 89.9 |

applications. In this study, we propose an annotation-efficient deep-learning framework, AIDE, for medical image segmentation network learning with imperfect datasets to address three challenges: SSL, UDA, and NLL. Our target is not to build a more sophisticated model for fully supervised learning, but to build a framework that can work properly without sufficient labeled data, so as to alleviate the reliance on the time-consuming and expensive manual annotations when applying AI to medical imaging. Extensive experiments have been conducted with open datasets. The results show that AIDE performs better than conventional fully supervised models under the same training conditions and even comparable to models trained with corresponding perfect datasets, thereby validating the effectiveness of AIDE. Additional experiments on clinical breast tumor segmentation datasets from three medical centers further prove the robustness and generalization ability of AIDE when processing real clinical data. On all three independent datasets, AIDE generates satisfactory segmentation results by utilizing only 10% of the training annotations, which indicates that AIDE can alleviate radiologists' manual efforts compared to conventional fully supervised DNNs.

Recent deep-learning works have presented encouraging performance in handling certain types of imperfect datasets in isolation[41–45] but have not yet shown general applicability by addressing all three types. Methods such as data augmentation[41,46], transfer learning[42,43], semi-supervised learning[44,47], and self-supervised learning[48,49] have been extensively investigated to handle cases with limited training annotations or no target domain annotations. By contrast, much less attention has been given to noisy label learning in medical imaging[16,50]. Most existing studies concentrate on designing new loss weighting strategies[51,52] or new loss functions[53]. Since various issues with the datasets may exist in reality, the effectiveness of these methods is compromised[22,26]. The available methods most closely related to ours are pseudo-labels[32] and co-teaching[27]. Compared to existing pseudo-label studies that update the pseudo-labels of all unlabeled data simultaneously during network learning[44,54,55], label updating in AIDE is conducted in an orderly (label updating is conducted according to the calculated similarities between the temporal network predictions and noisy labels in defined training epochs) and selective (only a defined percentage of noisy labels are updated) manner according to an observed small loss criterion, which has also been noted and confirmed for natural images[56]. Moreover, AIDE trains the model with the segmentation loss on filtered labels to avoid the negative effects of highly noisy labels, and an additional consistency loss is introduced to enforce consistent predictions with augmented inputs for the remaining suspected highly noisy data. AIDE also differs from the two-model co-teaching method[27,28]: AIDE generates pseudo-labels for suspected low-quality annotated data in an effective and progressive manner. During network training, a hyper-parameter that increases from 0 to 1 within defined initial training epochs is introduced. This hyper-parameter controls the

contribution of the consistency loss; thus, the consistency loss becomes increasingly important for network optimization in the training process. Additionally, the local label-filtering and global label-correction design progressively places more emphasis on pseudo-labels. These properties of AIDE contribute to the full utilization of valuable medical image data and improve the model performance.

Constructing large datasets with high-quality annotations is particularly challenging in medical imaging. The process of medical image annotation, especially dense annotation for image segmentation, is highly resource intensive. Different from natural images, only domain experts have the knowledge to annotate medical images. In some cases, such as our brain tumor segmentation task, large variations exist between experts (Fig. 1c), which raises the necessity of multiple annotators to achieve consensus annotations[17]. Employing a large number of domain experts to annotate large medical image datasets requires massive financial and logistical resources that are difficult to obtain in many applications. Our proposed AIDE has the potential to solve these issues effectively. It achieves promising performance by utilizing only 10% of the training annotations used by its fully supervised counterparts. Nevertheless, label-free or unsupervised learning that can eliminate manual annotation entirely is more appealing[20,57]. We are still working on improving the methodology to further reduce the reliance on annotations. In our following study, we will seek to address the challenge of unsupervised deep learning for large-scale and automatic medical image segmentation. Furthermore, although only medical image segmentation is considered in this work, AIDE might be applicable to other medical image analysis tasks, for example, image classification. The flexibility of AIDE in this perspective will also be evaluated in the future work.

In summary, we have analyzed and empirically demonstrated that our proposed framework, AIDE, can achieve accurate medical image segmentation to address the three challenges of semi-supervised learning, unsupervised domain adaptation, and noisy label learning. Therefore, AIDE provides another perspective for DNNs to handle imperfect training datasets. In the real-life case study, compared to conventional DNN training, AIDE can save almost 90% of the manual effort required to annotate the training data. With further development and clinical trials, such a ten-fold improvement in efficiency in utilizing expert labels is expected to promote a wide range of biomedical applications. Our proposed framework has the potential to improve the medical image diagnosis workflow in an efficient manner and at a low cost.

## Methods

**Public datasets**. Abdomen MR images from the CHAOS challenge are adopted[33,58,59] (Fig. 1c). These images were acquired with a 1.5 T Philips MRI system. The image matrix size is $256 \times 256$, with an inplane resolution from 1.36 to 1.89 mm. The slice thickness is between 5.5 and 9 mm. One 3D image contains 26–50 slices. Although multi-parametric MR images are provided, they are not registered and are thus difficult to utilize as multimodal inputs for the segmentation task. We utilize the T1-DUAL images. For each patient, there is one inphase image and one outphase image, which can be treated as multimodal inputs. In total, the challenge provides data from 40 patients, 20 cases (647 T1-DUAL images, 647 samples) with high-quality annotations and 20 cases (653 T1-DUAL images, 653 samples) without annotations. According to the challenge, image annotation was performed by two teams. Each team included a radiology expert and an experienced medical image processing scientist. Then, a third radiology expert and another medical imaging scientist inspected the labels, which were further fine-tuned according to discussions between annotators and controllers. For our setting of SSL, only the label of one randomly selected case (30 labeled image samples) is utilized, and pseudo-labels for the remaining data (29 cases including 9 randomly selected cases from the provided labeled cases and 20 unlabeled cases) are generated by a network trained with the labeled samples. Model testing is performed on the remaining ten labeled cases.

T2-weighted MR images of the prostate from two challenges (Fig. 1c), NCI-ISBI 2013[60] and PROMISE12[34], are utilized. NCI-ISBI 2013 consists of three groups of

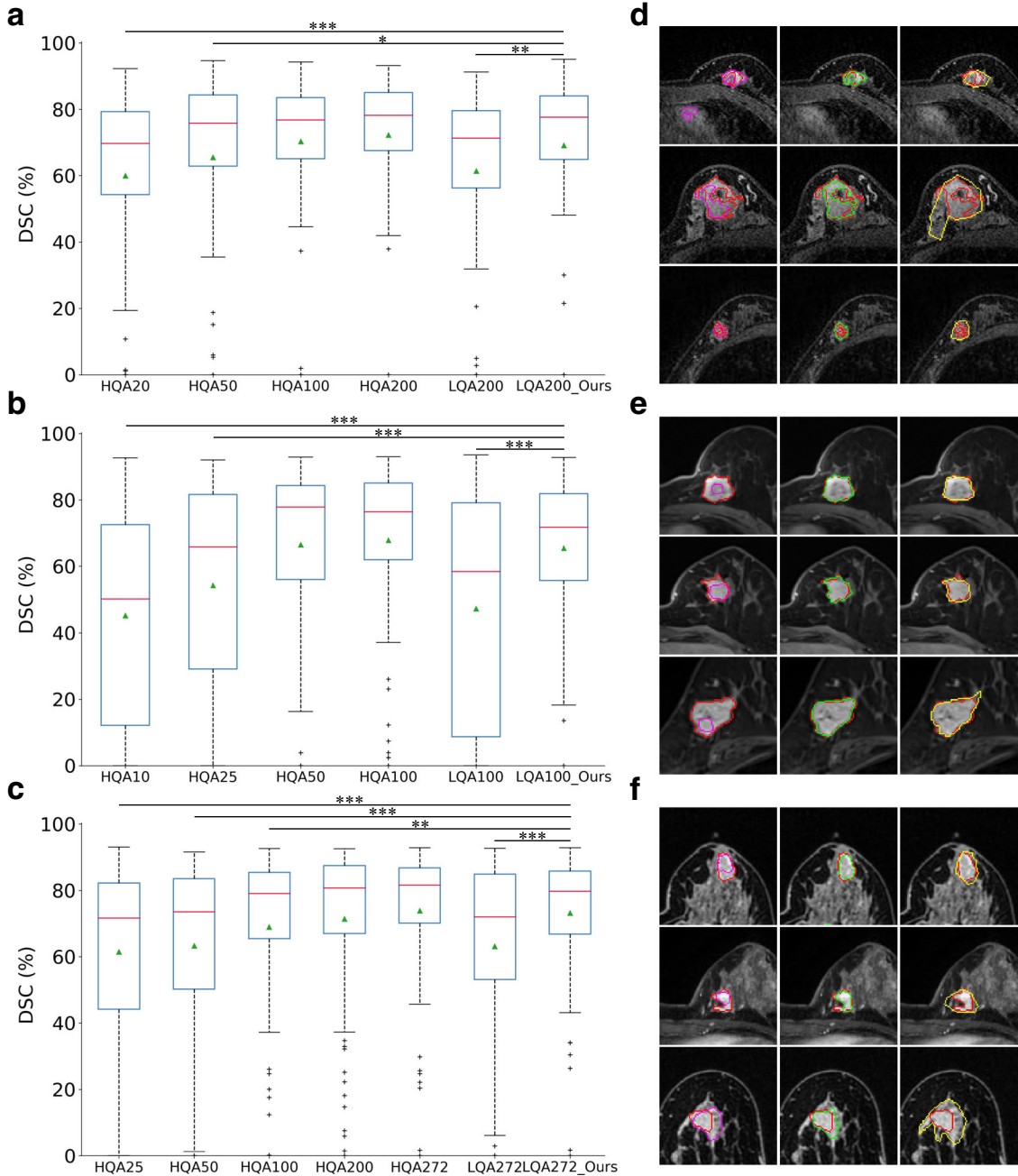

**Fig. 4 Results of breast tumor segmentation. a–c** Results on GGH, GPPH, and HPPH datasets, respectively. LQA and HQA indicate low- and high-quality annotations. The numbers after HQA refer to the annotations utilized to train the models. For LQA, the numbers indicate that we utilize the respective fewest HQA data with the remaining annotations generated by the pretrained models. For example, LQA200 in (**a**) means 20 high-quality labeled and 180 low-quality labeled data are utilized. Data are represented as box plots. The central red lines indicate median DSC values, green triangles the average DSC values, boxes the interquartile range, whiskers the smallest and largest values, and data points (+) outliers. * indicates a significant difference between the corresponding experiments, with ***$P \leq 0.001$, **$P \leq 0.005$, and *$P \leq 0.05$ (two-sided paired $t$ test, $n = 100$ independent patient cases). **a** Between HQA20 and LQA200_Ours, $P = 0.0002$; between HQA50 and LQA200_Ours, $P = 0.0139$; between HQA100 and LQA200_Ours, $P = 0.3119$; between HQA200 and LQA200_Ours, $P = 0.0608$; between LQA200 and LQA200_Ours, $P = 0.0017$. **b** Between HQA10 and LQA100_Ours, $P < 0.0001$; between HQA25 and LQA100_Ours, $P = 0.0002$; between HQA50 and LQA100_Ours, $P = 0.7602$; between HQA100 and LQA100_Ours, $P = 0.2927$; between LQA100 and LQA100_Ours, $P < 0.0001$. **c** Between HQA25 and LQA272_Ours, $P < 0.0001$; between HQA50 and LQA272_Ours, $P < 0.0001$; between HQA100 and LQA272_Ours, $P = 0.0075$; between HQA200 and LQA272_Ours, $P = 0.2925$; between HQA272 and LQA272_Ours, $P = 0.6545$; between LQA272 and LQA272_Ours, $P < 0.0001$. **d–f** Visualizations of segmentation maps on the three datasets. The three columns correspond to the results of LQA, LQA_Ours, and independent radiologists. Red contours indicate the high-quality annotations. Magenta, green, and yellow contours are the results of LQA, LQA_Ours, and the independent radiologists.

data: the training set (60 cases), the test set (10 cases), and the leaderboard set (10 cases). All are provided with high-quality annotations. Annotation was performed by two groups separately, one with two experts and the other with three experts. We combine the test set and leaderboard set to form our enlarged test set (20 cases) and divide the dataset into two datasets according to the data acquisition sites to form two domain samples. The domain 1 dataset contains training and testing data collected with 1.5 T MRI systems from Boston Medical Center (30 training cases and 10 test cases), and the domain 2 dataset contains data collected with 3.0 T MRI systems from Radboud University Nijmegen Medical Centre (30 training cases and 10 test cases). PROMISE12 provides data from 50 patients with high-quality annotations. Annotation was conducted by an experienced reader and then checked by a second expert. These samples were collected from different centers using different MRI machines with different acquisition protocols. Deleting 13 cases that are the same as those in domain 2, 37 cases collected from Haukeland University Hospital (12 cases), Beth Israel Deaconess Medical Center (12 cases), and University College London (13 cases) are obtained. Therefore, PROMISE12 can be treated as a combined dataset from different domains and is referred to as the domain 3 dataset in our experiments. Of the 37 cases, 27 are randomly selected as the training set, and the remaining 10 form the test set. UDA is constructed via model learning with labeled training data from the source domain and unlabeled training data from the target domain. Pseudo-labels are generated for the target domain training data, and the high-quality labeled source domain training data and low-quality noisily labeled target domain training data form the combined dataset to facilitate the domain transfer of the models. Performance is evaluated on the target domain testing data.

The QUBIQ challenge provides four datasets with annotations from multiple experts, including a prostate image dataset with 55 cases (6 sets of annotations), a brain growth image dataset with 39 cases (7 sets of annotations), a brain tumor image dataset with 32 cases (3 sets of annotations), and a kidney image dataset with 24 cases (3 sets of annotations). In total, there are seven binary segmentation tasks: two for prostate segmentation, one for brain growth segmentation, three for brain tumor segmentation, and one for kidney segmentation. In this study, we conduct four tasks (Fig. 1c), one for each dataset, to investigate the effects of our proposed AIDE framework. Following the challenge, 48 cases of prostate data, 34 cases of brain growth data, 28 cases of brain tumor data, and 20 cases of kidney data are utilized as the training data. The remaining are used as testing data. Each set of annotations is considered a noisy label set for model training. As defined by the challenge, performance characterization of models is conducted by comparing the predictions (continuous values in [0, 1]) to the continuous ground-truth labels (obtained by averaging multiple experts' annotations) by thresholding the continuous labels at different probability levels (0.1, 0.2, … 0.8, 0.9). Then, the DSCs for all thresholds are averaged to obtain the final metrics.

**Clinical datasets**. This retrospective study was approved by the institutional review board of each participating hospital with the written informed consent requirement waived. All patient records were de-identified before analysis and were reviewed by the institutional review boards to guarantee no potential risk to patients. The researchers who conducted the segmentation tasks have no link to the patients to prevent any possible breach of confidentiality.

Dynamic contrast-enhanced breast MR data from three medical centers, Guangdong General Hospital (GGH), Guizhou Provincial People's Hospital (GPPH), and Henan Provincial People's Hospital (HPPH), were investigated (Fig. 1d). FDA-approved fully features PACS viewer was used to collect the breast image data. GGH provided data for 300 patients, GPPH provided data for 200 patients, and HPPH provided data for 372 patients. For each dataset, three experienced radiologists with more than 10 years of experience provided the image annotations. Two radiologists delineated the breast tumor regions independently, and a third radiologist (the most experienced radiologist) checked the two sets of annotations and made the final decision. For the GGH dataset, breast tumor regions in central slices are annotated, leading to 300 image samples. For the GPPH dataset and HPPH dataset, 3D breast tumor annotations are provided. The GPPH dataset has 4902 annotated image samples, and the HPPH dataset contains 6650 image samples, resulting in a total of 872 MR data points (11,852 image samples) for our experiments. For each dataset, we randomly select 100 annotated patient cases as the respective hold-out test set and use the remaining cases as the training set.

**AIDE framework**. Figure 1 illustrates the overall AIDE framework. Task standardization is performed to transform SSL and UDA into NLL. For SSL, models are pretrained with the available limited annotated training data, and low-quality labels are generated for the remaining unlabeled training data. For UDA, models are pretrained with the source domain labeled training data, and low-quality labels are generated for the target domain unlabeled training data. Thus, the remaining issue becomes learning with possibly noisy annotations (NLL).

Details of the proposed self-correcting algorithm are presented in Algorithm 1 (Fig. 5). Two networks are learned in parallel to achieve cross-model self-correction of low-quality labels of imperfect datasets (Fig. 1b). The whole process can be divided into two major steps. The first is local label filtering. In each iteration, a defined percentage of training samples with relatively large segmentation losses is selected by the counterpart model as suspicious samples that have low-quality labels. For these samples, the final loss function is calculated as a weighted

summation of the segmentation loss and consistency loss. We choose the commonly utilized combination of Dice loss and cross-entropy loss as our segmentation loss in this work (Eq. (1)). The consistency loss is introduced to the network as a consistency regularization. Specifically, the means of the outputs of $K$ augmented inputs are calculated and are regarded as pseudo-labels after temperature sharpening[61,62]. Consistency loss, which is implemented as the mean square error (MSE) loss (Eq. (2)), is calculated between the network outputs of suspected noisy samples and the corresponding pseudo-labels. The samples with smaller segmentation losses are expected to be accurately labeled samples, and only the segmentation loss is calculated. The network parameters are updated according to the respective losses.

$$L_{seg}(y, y') = L_{Dice}(y, y') + \alpha \cdot L_{CE}(y, y') = \left(1 - \frac{2 \cdot \sum_{i=1}^{N} y'_i \cdot y_i + \varepsilon}{\sum_{i=1}^{N} y'_i + \sum_{i=1}^{N} y_i + \varepsilon}\right) - \frac{\alpha}{N} \left(\sum_{i=1}^{N} y_i \log(y'_i) + (1 - y_i) \log(1 - y'_i)\right) \quad (1)$$

$$L_{cor}(\hat{y}, y') = \frac{1}{2N} \sum_{i=1}^{N} \|\hat{y}_i - y'_i\|^2 \quad (2)$$

where $y'$ is the prediction of the network, $y$ is the reference, $\hat{y}$ is the temperature-sharpened local pseudo-labels generated from the augmented inputs, $N$ refers to the total number of pixels in the image, $\alpha$ is a constant to balance the different losses (we set $\alpha = 1$ in all experiments unless otherwise specified), and $\varepsilon$ is a constant to ensure numerical stability (we set $\varepsilon = 1.0$).

The second step is global label correction. After each epoch, the DSCs of the whole training set are calculated and ranked. A defined percentage of samples (25% in our experiments) with the smallest DSCs are selected, and their labels are updated when certain criteria are met. In this work, we update the labels if the training epoch number is smaller than the defined warm-up epoch number and every ten epochs thereafter.

The rationale for our designed filtering and correction steps is related to the frequently observed network memorization pattern for natural image analysis: deep neural networks tend to learn simple patterns before memorizing all samples and real data examples are easier to fit than examples with noisy labels[56,63,64]. Here, we find a similar network memorization pattern for medical image segmentation. By training a neural network utilizing the conventional fully supervised learning method with noisy labels, an overall positive correlation between the DSCs calculated by comparing network outputs to the noisy labels and the DSCs calculated by comparing the noisy labels to the high-quality labels exists for the training data during the initial training epochs (Supplementary Fig. 11a). In other words, within the considered training period, the network cannot successfully memorize those samples that contain large label noise but can learn the patterns of the samples that contain low label noise. On the other hand, all samples are well memorized when the network converges (Supplementary Fig. 11b). The noisy label updating schedule is designed accordingly—the suspected noisy labels are updated if the training epoch number is smaller than the defined warm-up epoch number and every ten epochs thereafter. The first criterion is based on the abovementioned observed network memorization behavior, and ablation studies indicate that the model performance is insensitive to the value of the warm-up epoch number in a large range (Supplementary Table 2). The second criterion is implemented because after the initial training epochs, the performances of the networks become relatively stable, and there is no need to update the labels frequently. Therefore, in our experiments, we update the labels every ten training epochs.

**Segmentation network architecture**. A simple, general, and robust network architecture (U-Net) is adopted to process the different image inputs[29]. The architecture employs a classical encoder−decoder structure with an encoder stream to downsample the image resolution and extract image features and a decoder stream to recover the image resolution and generate segmentation outputs[29]. Skip connections are included to improve the localization accuracy. Since the depth resolution is much lower than the inplane resolution for the liver and prostate segmentation datasets and the noisy datasets contain only 2D samples, all our experiments proceed in a 2D fashion. The 3D extension is straightforward but requires considerably more effort to construct suitable implementation datasets, which is out of the scope of this study.

Similar encoder−decoder network architectures are utilized to handle single- and multimodal image datasets. During encoding, input images pass through five downsampling blocks sequentially with four max-pooling operations in between to extract multilayer image features. For multimodal inputs, multiple downsampling streams are utilized to extract features from the different modalities[30,65]. The extracted features are combined in a multilayer fusion manner. The features from all downsampling blocks are correspondingly concatenated to fuse information from different modalities. During decoding, the extracted image features pass through four upsampling blocks, each followed by a bilinear upsampling operation, to generate segmentation outputs with the same size as the inputs. Downsampling and upsampling blocks have similar components: two convolutional layers with a $3 \times 3$ convolution, batch normalization, and ReLU activation. The low-level features of the encoder are introduced to the decoder through skip connections and feature concatenation. Finally, a $1 \times 1$ convolution is appended to generate two

---

**Algorithm 1 | Cross-Model Self-Correcting Mechanism of** AIDE.

1: **Input** Net1 (N1 with parameter $w_1$) and Net2 (N2 with parameter $w_2$), learning rate lr, training set D, batch size B, number of augmentations K, Sharpening temperature T, Segmentation loss function $L_{seg}$, Consistency loss function $L_{con}$, Loss weight $\lambda$, Label update ratio R, Label update criterion U, Warm-up epoch $q_w$, and total epoch Q

2: **for** q = 1, 2, … Q **do**

3: **Shuffle** training set D into |D|/B min batches;

4: **for** n = 1 to |D|/B **do**

5: **Fetch** batch inputs d = (($x_b$, $y_{b1}$, $y_{b2}$); b∈(1, …, B)) from D

 ($y_{b1}$, $y_{b2}$: references for N1 and N2, initialized to be the low-quality annotations);

6: **for** k = 1 **to** K **do**

7: **Obtain** $\hat{x}_{b,k}$ = Augment($x_b$)

8: **end for**

9: $\bar{y}_{b1} = \frac{1}{K}\sum_k (N_1(y \mid \hat{x}_{b,k}; w_1))$

10: $\hat{y}_{b1}$ = Sharpen($\bar{y}_{b1}$, $T$)

11: $(\hat{y}_{b1,m} = \frac{\exp(\frac{\bar{y}_{b1,m}}{T})}{\sum_n \exp(\frac{\bar{y}_{b1,n}}{T})}$ with $m$ and $n$ referring to the categories)

12: $\bar{y}_{b2} = \frac{1}{K}\sum_k (N_2(y \mid \hat{x}_{b,k}; w_2))$

13: $\hat{y}_{b2}$ = Sharpen($\bar{y}_{b2}$, $T$)

14: **Obtain** $d_{s1}$ = argmin $_{d' \in d : |d'| \leq 0.5B}$ $L_{seg}$(d'; $w_1$) //select 50% small loss samples in the batch

15: **Obtain** $d_{s2}$ = argmin $_{d' \in d : |d'| \leq 0.5B}$ $L_{seg}$(d'; $w_2$) //select 50% small loss samples in the batch

16: **Obtain** L1 = $L_{seg}(N_1(y \mid d_{s2}; w_1), y_{d_{s2}})$ + (1.0 - $\lambda_q$) $L_{seg}(N_1(y \mid d_{l2}; w_1), y_{d_{l2}})$ +

 $\lambda_q L_{con}(N_1(y \mid d_{l2}; w_1), \hat{y}_{d_{l2}})$, $d_{l1}$ = d / $d_{s1}$

17: **Obtain** L2 = $L_{seg}(N_2(y \mid d_{s1}; w_2), y_{d_{s1}})$ + (1.0 - $\lambda_q$) $L_{seg}(N_2(y \mid d_{l1}; w_2), y_{d_{l1}})$ +

 $\lambda_q L_{con}(N_2(y \mid d_{l1}; w_2), \hat{y}_{d_{l1}})$, $d_{l2}$ = d / $d_{s2}$

18: **Update** $w_1$ = $w_1$ − lr $\nabla$L1

19: **Update** $w_2$ = $w_2$ − lr $\nabla$L2

20: **end for**

21: **Update** $\lambda_q$ = min{$\lambda$ * (q / $q_w$)$^2$, $\lambda$}, $\lambda$=1.0

22: **Obtain** $D_{l1}$ = argmin $_{D' \in D : |D'| \leq R|D|}$ Dice(D'; $w_1$) //select R% small loss cases in the whole training set

23: **Obtain** $D_{l2}$ = argmin $_{D' \in D : |D'| \leq R|D|}$ Dice(D'; $w_2$) //select R% small loss cases in the whole training set

24: **if** U **do**

25: **Update** $y_{b2}$ = $N_1$(y | $D_{l1}$; $w_1$) if $D_{l1}$ is not high-quality labeled

26: **Update** $y_{b1}$ = $N_2$(y | $D_{l2}$; $w_2$) if $D_{l2}$ is not high-quality labeled

27: **Obtain** $DSC_{train}$ = mean(Dice(D; $w_1$), Dice(D; $w_2$))

28: **end if**

29: **end for**

30: **Output** $w_1$ and $w_2$ with the highest $DSC_{train}$

---

**Fig. 5 Pseudo-code of AIDE.** The inputs required for the model training, the overall training process with the proposed cross-model self-correcting mechanism, and the optimized model to be saved are included.

features corresponding to the background and target segmentation maps, and a softmax activation is included to generate the segmentation probability maps.

**Evaluation metrics**. Different metrics can be used to characterize the segmentation results. For our experiments, we choose the commonly utilized Dice score (Dice similarity coefficient, DSC), relative area/volume difference (RAVD), average symmetric surface distance (ASSD), and maximum symmetric surface distance (MSSD).

$$DSC = \frac{2TP}{2TP + FP + FN} \tag{3}$$

$$RAVD = \frac{FP - FN}{TP + FN} \tag{4}$$

$$ASSD = \frac{1}{|S(y')| + |S(y)|} \left( \sum_{a \in S(y')} \min_{b \in S(y)} ||a - b|| + \sum_{b \in S(y)} \min_{a \in S(y')} ||b - a|| \right) \tag{5}$$

$$MSSD = \max \left( \max_{a \in S(y')} \min_{b \in S(y)} ||a - b||, \max_{b \in S(y)} \min_{a \in S(y')} ||b - a|| \right) \tag{6}$$

where TP, FP, and FN refer to true positive predictions, false positive predictions, and false negative predictions, respectively. $S(y')$ and $S(y)$ indicate the boundary points on the predicted segmentations and reference segmentations.

Significant differences between the different experiments and between the model results and human annotations are determined using two-sided paired $t$ tests, with $P \leq 0.05$.

**Statistics and reproducibility**. The code used for training the deep-learning models are made publicly available for the reproducibility purpose. Statistical analysis has been given as well. Specifically, we run the code three times with different random initializations for the CHAOS dataset. For the domain adaptation task on prostate segmentation, six independent experiments were performed. On the QUBIQ datasets, we repeated 6, 7, 3, and 3 times respectively for the four different subtasks according to their dataset properties. For our breast datasets, data from three hospitals were utilized. So the experiments were performed independently for three times.

**Reporting summary**. Further information on research design is available in the Nature Research Reporting Summary linked to this article.

## Data availability

The raw image data and relevant information of the utilized open datasets are accessible from the respective official websites of the challenges (CHAOS: https://chaos.grand-challenge.org/, NCI-ISBI 2013: https://wiki.cancerimagingarchive.net/display/Public/NCI-ISBI +2013+Challenge+-+Automated+Segmentation+of+Prostate+Structures, PROMISE12: https://promise12.grand-challenge.org/, and QUBIQ: https://qubiq.grand-challenge.org/) through standard procedures. The clinical breast data were collected by the hospitals in de-identified format. Owing to patient-privacy considerations, they are not publicly available. All requests for academic use of in-house raw and analyzed data can be addressed to the corresponding authors. All requests will be promptly reviewed within 10 working days to determine whether the request is subject to any intellectual property or patient-confidentiality obligations, will be processed in concordance with institutional and departmental guidelines, and will require a material transfer agreement (available at https://github.com/lich0031/AIDE). Source data are provided with this paper.

## Code availability

The code used for training the deep-learning models and models used in this study are made publicly available[66]. Implementation of our work is based on PyTorch with necessary publicly available packages, including numpy, pandas, PIL, skimage, and SimpleITK.

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

## Acknowledgements

This work was partly supported by Scientific and Technical Innovation 2030-"New Generation Artificial Intelligence" Project (2020AAA0104100, 2020AAA0104105), the National Natural Science Foundation of China (61871371, 81830056), Key-Area Research and Development Program of Guangdong Province (2018B010109009), the Basic Research Program of Shenzhen (JCYJ20180507182400762), Youth Innovation Promotion Association Program of the Chinese Academy of Sciences (2019351), National Key R&D Program of China (2017YFE0103600), National Natural Science Foundation of China (81720108021), Zhongyuan Thousand Talents Plan Project (ZYQR201810117), Zhengzhou Collaborative Innovation Major Project (20XTZX05015). We thank all the participants and staff of the four challenges, CHAOS, NCI-ISBI 2013, PROMISE12, and QUBIQ. We thank Professor H. Peng for discussions.

## Author contributions

S.W. and C.L. proposed the idea and initialized the project and the collaboration. C.L. and S.W. developed and implemented the framework. R.W., Z.L., M.W., H.T., and Xinfeng L. collected the breast data. R.W., Z.L., M.W., Y.W., and R.Y. provided the manual annotations. R.W., Z.L., M.W., H.T., Xinfeng L., Y.W., R.Y., and Xin L. contributed clinical expertise. C.L. and H.S. analyzed the results and plotted the figures. C.L., H.S., J.C., H. Zhou, I.B.A., and S.W. wrote the manuscript. S.W. and H. Zheng supervised the project. All authors read and contributed to revision and approved the manuscript.

## Competing interests

The authors declare no competing interests.
