## [Peer Review File · Nature Communications]

Reviewers' Comments:

Reviewer #1:

Remarks to the Author:

This paper proposes a solution to the case where there are not enough accurate annotations, which is a very important problem in medical image AI/ML. This paper proposes that we could be able to handle imperfect training datasets through AIDE(Annotation-Efficient Deep Learning).

This paper proposes a solution to the case where there are not enough accurate annotations, which is an essential problem in medical image AI/ML.

This paper proposes that we could handle imperfect training datasets through AIDE(Annotation-Efficient Deep Learning).

This paper describes that much-improved performance can be provided even with noisy data sets compared to the conventional fully supervised model.

While this paper captures a critical issue, it is unfortunate that it does not fully consider some of the more critical matters.

Even if it is an algorithm that works well for a general natural image, it does not work well for a medical image. This is because available natural images collect very different photos, and segmentation or classification in medical images primarily distinguishes slight differences from very similar images.

That is why it is essential to understand and define the problem well. It is difficult for one or two experienced doctors to determine the ground truth in a medical image.

This is because, in many cases, slight differences in detail can lead to many discrepancies.

In particular, when measuring organ segmentation or disease segmentation with various metrics, it is challenging to determine the ground truth. In particular, depending on the boundary drawn by experts, the algorithm's performance quickly differs by more than 10%.

In other words, it is not possible to know whether the performance of the algorithm is improved or whether the ground truth is biased. This problem is mainly unknown when data from various sources are imported and applied to each other or combined for learning and testing.

It is not surprising that an algorithm that learns about 10% of random data by combining multiple data sets is better than training an algorithm from one data source and applying it to another data source.

And, although it was said that a noisy label was used, there was no information on how quantitatively noisy data is okay to use. This is because how much missing or incorrect annotations affect the system's performance significantly.

And, in figure 4, using clinical data, it can be seen that the performance does not improve and does not deteriorate even when the proposed method is used. And the overall performance is about 75%.

It is a performance that seriously lacks safety guidelines for medical use where malpractice or mistakes can be disastrous to patients.

In this paper, many essential parts are not clearly explained.

For example, line 125-126 "After each epoch, the labels of the whole training set are analyzed and those that have low similarity with the network predictions are updated if a pre-defined updating criterion is met."

=>Here, the most important "low similarity" and "pre-defined updating criterion" require precise explanation and description.

Line 127-128 "With AIDE, the networks are forced to concentrate more on the image contents instead of extracting image features guided only by the annotations"

=>There is no explanation as to how AIDE concentrates on the contents of an image, which should be explained in this paper.

line 145-146: "In other words, the networks are expected to have small vibrations and those samples that lead to large vibrations are considered low-quality annotated samples that should be corrected."

The authors need to explain how this is implemented clearly so that we could be able to judge the value of this study.

line 147 "Direct label correction to replace the old labels offers the network a self-evolving capability"

This is also conceptual content and requires a scientific and logical step-by-step explanation.

lines 228-229 "With our proposed method, the model can self-adjust and update the training annotations to correct
229 the possible observer biases and generate more satisfactory segmentation results"

This, too, needs a detailed explanation but ends with an argument. Here, it is necessary to explain and show how the ground truth is defined clearly.

Lines 257-260 "GGH", "GPPH", and "HPPH" should be spelled out here.

I am very curious about how line 301 "orderly and selective manner" is, and it should be explained in detail and clearly.

Line 306 "in an effective and progressive manner," has no scientific explanation or logical interpretation in the paper.

line 428 "Here we show that the same network memorization pattern exists for medical image segmentation."

No explanation or evidence appears in the related medical image at all in this paper.

Reviewer #2:

Remarks to the Author:

This work introduces AIDE, a framework for automatic medical image segmentation which accounts for 3 types of incomplete annotation scenarios, including semi-supervised learning (SSL, where part of the training dataset is unlabelled), unsupervised domain adaptation (UDA, where the target domain is unlabelled) and noisy label learning (NLL, where annotations are noisy). The authors propose to first convert the SSL and UDA cases to a NLL scenario by means of a task standardization procedure. Such task standardization procedure is in fact a pseudo-label generation strategy, where a CNN is trained using the fully annotated data and labels are simply inferred for the unlabelled images, generating the 'noisy' annotations. Such labels are then treated as a NLL scenario. The AIDE framework tackles NLL by leveraging a cross-model co-optimization procedure, where 2 networks are jointly trained. During the training procedure, noisy labels are filtered out and corrected following the so-called local and global cascaded steps.

The method is experimentally validated in the three incomplete annotation scenarios using images corresponding to a variety of publicly available datasets (CHAOS, PROMISE12, NCI-ISBI 2013 and

QUBIQ) and one private clinical dataset, all of which are treated as 2D images. Results suggest that the proposed framework is effective at deadline with NLL scenarios.

Overall, the paper is well written, code is available and I think it makes an interesting effort trying to unify three different but related problems, under the same umbrella. Moreover, the experimental results in a variety of datasets suggest that the proposed framework actually alleviates the problem of limited annotations in several scenarios. I believe a work like this would be of interest for the machine learning, computer vision and medical imaging communities. However, there are several weak points related to novelty and lack of presentation clarity that make me think that this paper may not be ready for publication in a journal like Nature Communications. Let me detail these points and provide some constructive suggestions:

== Novelty ==

Even though I like the effort made by the authors to unify the 3 problems (SSL, UDA and NLL) under the same umbrella, I wonder if the incremental methodological novelty with respect to existing co-training and pseudo-label methods is enough for a journal like Nature Communications.

== Missing "theoretical" evaluations ==

In line 87 the authors mentioned that "theoretical analysis is performed to evaluate the effectiveness of AIDE...". However, I do not see any theoretical analysis, proof or demonstration supporting the claims made by the authors. On the contrary, most of the validation seems to be experimental. Am I missing something here?

== SOTA discussion ==

Since most of the public datasets used by the authors are associated with challenges, I would have expected a discussion positioning the accuracy obtained by AIDE with respect to the state of the art in every problem, not in the tables but at least in the Discussion section. How far are the results obtained from the winners of such competitions for example? Or from more modern methods which used the same datasets?

== Pseudo-label baseline ==

In table 1, it is not clear to me what is the difference between the pseudo-label baseline and the settings like S10, where AIDE is "No" and you use LQ annotations. As far as I understand, LQAs are generated as pseudo-labels using the HQ labels for training. Thus, I do not see the difference with a vanilla pseudo-label approach. I may be missing something here because numerical results are different, but please clarify this in the manuscript.

Also, why is the pseudo-label baseline only used when comparing with SSL, but not as a UDA baseline in Figure 3? Couldn't it be applied in that setting?

== Clarity of the presentation ==

In my opinion, the main methodological novelty of this work is related to Algorithm 1. However, there are aspects of this algorithm that are not clear to me. In particular:

--- Does the algorithm enforce in any way an explicit differentiation in case of UDA and SSL scenarios between the pseudo-labels and the high quality ones? As far as I understand, this is not explicitly enforced because accurately annotated samples are selected in lines 15 and 16 of the Algo1, by looking at the segmentation loss. If this is not explicitly enforced, aren't you losing the possibility of taking advantage of those labels that you know are of high quality? I may be missing something here. Please clarify this point.

--- Also related to lines 15 and 16, it is not clear to me what are the arguments of the argmin. In particular, what is $d \in \{d: |d'| < 0.5B\}$?

--- In line 411, L_{seg} is defined as a metric which takes 2 segmentation masks as arguments. However, when used in Algo1 (lines 15, 16, 17, 18), it receives as a parameter a batch d' and the weights w_1 or w_2 . Please clarify if you are overloading the operator and, in that case, what does it exactly mean?

--- What is q_t in line 22 of Algorithm 1?

--- Indentation of algorithm 1 does not correspond with the beginning/end of the for loops. This should be corrected to ease understanding. Sentences which are at the same nesting level should also be at the same indentation level.

== Figure 3 ==

Figure 3 is a bit confusing. First, intuitively one would expect that red color relates to better performance while blue relates to lower performance. However, figures a to d include a mix of measures, some of which the lower the better (e.g. ASSD), and other in which the higher the better (e.g. Dice), and all of them use the same color-bar scheme. I would adjust the color bars so that blue means better performance and red means worse, or vice versa, in all the cases. Otherwise it's a bit confusing. Also, please add a title to every matrix indicating what metric it is referring to. That will help the readers and simplify their work, avoiding the need to check the figure caption all the time.

== Rationale of the proposed label filtering method ==

In line 423, the authors comment that "The rationale for our filtering and correction steps is that we observed when training data containing noisy annotations, there is a positive correlation between the DSCs calculated comparing network outputs to current noisy labels and the DSCs calculated comparing current noisy labels to the high-quality labels for the training set" and they refer to Figure S11 in the Sup Mat. I do not clearly see how this relates to the rationale of the proposed method. I would like the authors to discuss it more explicitly.

Reviewer #3:

Remarks to the Author:

The authors describe AIDE, a method for training deep segmentation models on medical images. Their method is similar to co-training, optimizing two models simultaneously and using them to generate pseudo-labels. This technique appears to be novel, and based on the results presented, training using AIDE appears to improve network performance for semi-supervised tasks, and data with noisy labels. The MR images used for segmentation in this study are all 2D, which is ok in the context of the paper. The claim that medical datasets lack annotation, and that annotation is costly is also well validated in the literature.

The relationship between transforming unsupervised domain adaptation (UDA) and semi supervised learning (SSL) into a noisy label learning (NLL) task is described in the methods section. This relationship is essential to the readability of the paper and should have been presented early in the introduction. For example, this relationship is not clear in figure 1.

Overall the descriptions of the figures require further work. For example, in figure 2 the difference between the panels with the white and black background are not described.

The S## (example S01) and Anno# descriptions of the experiments / training setups used in the work are very confusing to the reader. This notation could be improved to give some context to the experimental setup used without referencing table 1. Maybe the AIDE vs non AIDE training results could be color coded for ease of interpretability.

The figures references in the text seem to be an afterthought. The readability could be improved by referencing the specific figure sub panel more frequently (in the correct context).

Descriptions of how the ground truth was generated are lacking in the text. The authors compare the annotations of multiple radiologists, but do not describe how the ground truth was derived for the high-quality labeled training set (for example was one radiologist's annotation chosen?).

The authors use a self-created architecture for segmentation. The training done in the paper is

done from scratch. It is unclear how utilizing a previously trained more popular architecture (for example DeepLab) that has been pre-training on a natural image dataset may affect segmentation generalizability. Many works have shown that the use of pre-trained architectures for transfer learning produces very good results for medical image segmentation etc. It would be interesting to compare transfer learning to AIDE for segmentation.

To the same effect, the authors describe the AIDE technique as independent to the network architecture employed. Why was a custom architecture created instead of one already proven for segmentation?

The descriptions of the experimental protocols for training are not well described. The reader has to dig deeply into the results section to understand how training was performed. For example, with the UDA results in table 2, it is unclear if the testing dataset was used for training with pseudo-labels generated via AIDE.

The AIDE technique appears to be applicable to domains of medical image analysis beyond segmentation. This would enhance the appeal of the work to a greater audience.

Overall, this paper presents a novel framework for reducing the annotation burden for medical image segmentation via deep learning. In the reviewer's opinion, the readability of the work is not up to the standards of Nature Communications. The figures are not well incorporated into the write up, and the experimental setups are not apparent without close inspection and stipulation. The reviewer acknowledges that this is a complex topic, which may be hard to describe, but a hallmark of the Nature style is the readability by a greater audience.

AIDE: Annotation-efficient deep learning for automatic medical image segmentation

Response to referees letter

Responses to Reviewer #1

Comment 1: This paper proposes a solution to the case where there are not enough accurate annotations, which is a very important problem in medical image AI/ML. This paper proposes to handle imperfect training datasets through AIDE (Annotation-efficient deep learning). This paper describes that much-improved performance can be provided even with noisy data sets compared to the conventional fully supervised model.

Response: Many thanks for recognizing our efforts in addressing an essential issue about AI applications in medical imaging and for recognizing the encouraging performances we have obtained. We have carefully gone through all your comments, which has further helped us to improve the manuscript. We hope that our responses and the revised manuscript are able to address all your concerns.

Comment 2: While this paper captures a critical issue, it is unfortunate that it does not fully consider some of the more critical matters. Even if it is an algorithm that works well for a general natural image, it does not work well for a medical image. This is because available natural images collect very different photos, and segmentation or classification in medical images primarily distinguishes slight differences from very similar images. That is why it is essential to understand and define the problem well. It is difficult for one or two experienced doctors to determine the ground truth in a medical image. This is because, in many cases, slight differences in detail can lead to many discrepancies. In particular, when measuring organ segmentation or disease segmentation with various metrics, it is challenging to determine the ground truth. In particular, depending on the boundary drawn by experts, the algorithm's performance quickly differs by more than 10%. In other words, it is not possible to know whether the performance of the algorithm is improved or whether the ground truth is biased. This problem is mainly unknown when data from various sources are imported and applied to each other or combined for learning and testing. It is not surprising that an algorithm that learns about 10% of random data by combining multiple data sets is better than training an algorithm from one data source and applying it to another data source.

Response: Thanks a lot for recognizing that our work is trying to address a critical issue and for raising this inspiring question on the label issue. We agree with the reviewer that annotating medical images is difficult and different experts can give slightly different results depending on his/her own experience. Medical imaging-related tasks are indeed different from natural image processing. This is in fact one of our motivations to develop this method – building deep neural networks, which can concentrate more on the input image data instead of the provided labels to circumvent the possible biases or noises introduced by labels.

We also agree with the reviewer that the performance can differ depending on the boundaries drawn by experts. Characterizing the performance of our proposed method as well as literature-reported methods need ground-truth segmentation labels. That's why we implement our method on various publicly available datasets. These datasets were carefully curated with labels provided by summarizing the opinions of multiple experienced radiologists with quality controls.

We hope that by utilizing these widely accepted data and labels, we are evaluating our method fairly without severe ground-truth label biases. As shown by the experiments on the three public datasets, our method achieves better performance than the methods we compared with.

For the issue of the rationale of combining multiple data sets, we want to clarify that we are not only comparing our method trained with multiple datasets to those methods trained with single datasets (Table 2 in the manuscript), but also comparing our method to those methods trained with multiple datasets (Figure 3 in the manuscript); our method always performs better. With these achievements, we make the conclusion that the proposed method is effective for the unsupervised domain adaptation task we are tackling.

Also, since the public data are highly standardized, we employ the clinical data for breast tumor segmentation to evaluate the feasibility of our method for real-world applications. The ground-truth labels for these data are provided by three experienced radiologists with more than 10 years of experience in breast MR image interpretation. Two radiologists delineate the breast tumor regions independently, while a third radiologist (the most experienced one) checks the two sets of annotations and makes the final decision. We show that, although there is still room for improvement, our method gets a significantly higher DSC value ($P < 0.05$ with two-tailed paired t -tests) when compared to that achieved by the conventional fully-supervised method.

In the introduction of the revised manuscript, we have added more details: “In order to fairly evaluate the method without severe ground-truth label biases, we conduct extensive experiments on a variety of public datasets which have widely accepted data and labels.” (page 5 lines 94–96) and “Furthermore, to test the applicability of our method in real-world applications, three datasets for breast tumor segmentation (11,852 image samples of 872 patients) collected from three medical centers are experimented with. The annotations of these data are carefully curated by three experienced radiologists with more than 10 years of experience in breast MR image interpretation.” (page 5 lines 99–103).

Comment 3: Although it was said that a noisy label was used, there was no information on how quantitatively noisy data is okay to use. This is because how much missing or incorrect annotations affect the system's performance significantly.

Response: Sorry for the confusion. We have added more explanations and supplemented more experimental results to address your concern (page 9 lines 184–192 and lines 198 and 199).

Noisy segmentation labels can be measured in a pixel-based level (each mis-classified pixel is counted as one noisy case) or image-based level (each image is counted as one noisy case regardless of how many pixels in the image are mis-classified). We choose the latter in our experiments. Please be noted that, we treat *the labels generated by pre-trained models or provided by independent annotators without quality controls as noisy labels*. Our proposed AIDE framework can be applied to three scenarios, semi-supervised learning (SSL), unsupervised domain adaptation (UDA), and noisy label learning (NLL). The noise levels are different for different application scenarios. For UDA and NLL, the noisy label percentages are inherently 100%. Specifically, for UDA, all the labels for the target domain training data are noisy (labels are generated by models trained with the source domain labeled training data). NLL has 100% noisy labels caused by its task property (labels are given by individual annotators without quality controls). For these two tasks with 100% noisy labels, results show that our method is applicable with enhanced segmentation performance compared to the conventional fully-supervised learning

baselines as they rely on high-quality labels (Fig. 3 and Table 3 in the manuscript).

We investigate the influence of the number of noisy labels on segmentation performance systematically and comprehensively with SSL, for which low-quality labels (i.e. noisy labels) are generated by models trained with the limited annotated data. The combined training data (high-quality labeled data and low quality noisily labeled data) are then utilized for network optimization. Table R1 lists the results under different experimental conditions. For the fully-supervised learning baseline, noisy labels already affect their performance when the noise level is over 20% ($P = 0.0133$ from a paired t -test between the DSCs of Exp. 1 and Exp. 2 in Table R1). Compared to the baseline, our method consistently improves the performance (characterized by increased DSC and decreased RAVD) at different noise levels. Furthermore, promising performance is achieved by our method with 97% noisy labels (Exp. 14 in Table R1).

Table R1 Segmentation results of networks under different SSL settings									
Exp.	Noisy labels (%)	Train HQA	Train LQA	Total train	AIDE	DSC (%)	RAVD (%)	ASSD (mm)	MSSD (mm)
1	0	331	0	331	No	87.9	10.4	4.65	65.1
2	20.2	264	67	331	No	84.7	16.5	4.48	49.3
3	38.7	203	128	331	No	81.5	16.0	6.23	66.8
4	58.3	138	193	331	No	80.2	19.4	6.31	78.4
5	80.7	64	267	331	No	79.4	17.9	8.16	115.9
6	90.9	30	301	331	No	78.4	19.0	7.92	95.3
7	97.0	30	954	984	No	79.5	19.7	8.43	104.2
8	0	331	0	331	Yes	88.1	11.1	4.71	61.5
9	20.2	264	67	331	Yes	87.3	13.2	5.08	71.9
10	38.7	203	128	331	Yes	84.3	14.9	5.80	72.9
11	58.3	138	193	331	Yes	84.0	14.6	10.4	119.0
12	80.7	64	267	331	Yes	82.9	16.9	7.94	108.2
13	90.9	30	301	331	Yes	79.8	18.5	10.8	116.3
14	97.0	30	954	984	Yes	86.1	10.2	5.49	75.8

In summary, the proposed system is applicable for the two tasks of UDA and NLL with 100% noisy labels, and it achieves promising performance at high noise levels (such as the 97% noise level experimented with) for SSL.

We have put all these results and discussions in the revised manuscript (page 9 lines 184–192 and lines 198 and 199) and revised supplementary file (page 3 lines 84–96 in the supplementary information and Supplementary Table 1).

Comment 4: In figure 4, using clinical data, it can be seen that the performance does not improve and does not deteriorate even when the proposed method is used. And the overall performance is about 75%. It is a performance that seriously lacks safety guidelines for medical use where malpractice or mistakes can be disastrous to patients.

Response: We apologize for the lack of clarity here. To clarify this issue and address the reviewer’s concern, we have added more details and discussions in regard to two aspects (pages 13 and 14 lines 284–292 and pages 14 and 15 lines 309–322): 1) In comparison to the experiments

utilizing the conventional fully-supervised learning approach under the same experimental settings, our method improves the segmentation performance significantly, increasing the Dice similarity coefficient values by more than 7% (7.7% for GGH dataset, 18.2% for GPPH dataset, and 10.0% for HPPH dataset). Please kindly refer to the two-tailed paired t -tests in Fig. 4. It can be observed that there are significant differences (indicated by the stars in the figure) between the corresponding experiments with the same training data conditions ($P = 0.0017$ between LQA200 and LQA200_Ours in Fig. 4a, $P < 0.0001$ between LAQ100 and LQA100_Ours in Fig. 4b, and $P < 0.0001$ LQA272 and LQA272_Ours in Fig. 4c). 2) Our method can achieve the average radiologist-level performance that is comparable to those obtained by the state-of-the-art fully-supervised methods. In Fig. 4, comparable segmentation performances are obtained by our method (with 10% training labels, e.g. LQA200_Ours) and by fully-supervised models (with 100% training labels, e.g. HQA200). Further comparisons of the results achieved by our method to additional radiologists' annotations also indicated no significant difference. Furthermore, similar segmentation performance has been reported by state-of-the-art fully-supervised methods trained with hundreds of labeled patient cases. For example, Zhang et al. built a hierarchical convolutional neural network framework that contains three fully convolutional neural networks to segment breast tumors in DCE-MR images, and they achieved a mean DSC value of 72%¹. Qiao et al. reported a mean DSC value of 78% with 3D U-Net when single-phase DCE-MR images were utilized². Our method achieves a DSC of 73.1% on the HPPH dataset utilizing 25 labeled training cases and 247 unlabeled cases. Therefore, we can confidently state that our method is able to improve the segmentation performance in real clinical applications when limited annotations are provided. *We have made our code publicly available for reference.*

To make the paper more rigorous, we also added more discussions to highlight that our method is not to replace radiologists in the disease diagnosis or treatment planning workflow but to serve as an automated computer-aided system (pages 14 and 15 lines 313–315). It is emphasized that our proposed method is to serve as a tool in clinical practices to assist radiologists in achieving fast and reliable breast tumor segmentation (page 15 lines 319–322).

Comment 5: Line 125-126 "After each epoch, the labels of the whole training set are analyzed and those that have low similarity with the network predictions are updated if a pre-defined updating criterion is met." Here, the most important "low similarity" and "pre-defined updating criterion" require precise explanation and description.

Response: Following your advice, we have supplemented the details on page 23 in lines 497–501 in the revised manuscript to clarify the two descriptions. Similarities between labels and network predictions are measured by the Dice similarity coefficients (DSCs). A lower similarity means a smaller DSC, and we only update a defined percentage of training data labels ranked by the calculated similarity values. In this work, we update 25% of the labels for all experiments after each epoch if the pre-defined updating criterion is met. The pre-defined updating criterion is a specific schedule we designed. We only update the selected labels if the training epoch number is smaller than the defined warm-up epoch number and every 10 epochs thereafter. The warm-up epoch number in this work is set to 20. The total epoch number for network training is 100. The performance of our method is not sensitive to this warm-up epoch number as indicated by our ablation studies (Supplementary Table 2).

Comment 6: Line 127-128 "With AIDE, the networks are forced to concentrate more on the image contents instead of extracting image features guided only by the annotations". There is no explanation as to how AIDE concentrates on the contents of an image, which should be explained in this paper.

Response: Thanks for pointing out this issue. In the revised manuscript, we have explained how AIDE concentrates more on the image contents in section "Enhancement of AIDE compared to conventional fully-supervised learning" (pages 7 and 8 lines 152–156). For traditional fully-supervised learning, the network gets the inputs and generates the corresponding outputs. A loss is calculated between the outputs and the ground-truth labels, and network training is performed by minimizing this loss. Thus, the network parameter optimization process is largely determined by the ground-truth labels, and any biases in the labels can be transferred to the learned models³. With our method, we introduce a consistency loss between the model predictions of the inputs and averaged model predictions of augmented inputs for suspected noisily labeled data. This consistency loss implicitly enforces a data distillation regularization on the model and forces the model to learn latent features from inputs that are transformation (random rotation and flipping) invariant. In this way, the model can concentrate on the image content.

Comment 7: "In other words, the networks are expected to have small vibrations and those samples that lead to large vibrations are considered low-quality annotated samples that should be corrected." The authors need to explain how this is implemented clearly so that we could be able to judge the value of this study.

Response: Following your advice, we have supplemented additional implementation details (page 8 lines 158–161 and page 23 lines 497–501). At the end of each training epoch, we calculate the Dice similarity coefficients (DSCs) between the network predictions in the current epoch and the pseudo-labels generated in previous epochs for the noisily labeled data. We rank the samples according to the calculated DSC values and select the 25% of samples that have the lowest values for label updating every 10 epochs or if the epoch number is smaller than the defined warm-up epoch number. In this way, the pseudo-labels of samples corresponding to high consistencies in network predictions across different epochs are kept. Therefore, we have the statement "the networks are expected to have small variations" with variations referring to the changes in the network predictions. Please note that we change the word "vibrations" to "variations" in the revised manuscript. This design can be considered as a method for model ensemble. By integrating the results of networks at different training epochs, we try to find the robust patterns that the network predicts. Implicitly, we assume that noisy labels have a higher possibility of leading to different predictions at different network training stages since noisy labels, to a certain extent, contradict the information provided in the images. We have added these discussions to the revised manuscript (page 8 lines 158–161 and page 23 lines 497–501).

Comment 8: Line 147 "Direct label correction to replace the old labels offers the network a self-evolving capability". This is also conceptual content and requires a scientific and logical step-by-step explanation.

Response: Following your suggestion, we have provided more explanations (page 8 lines 162–164). Our intention of this self-evolving capability statement is to emphasize that the

pseudo-labels are generated by the model and are further utilized in turn to help the model training in the subsequent steps. Although two neural networks are utilized to prevent the error propagation in one network, the whole framework can be treated as a whole that can modulate itself without external help. Therefore, the framework works in way that attempts to find useful information from its own outputs, which fits the definition of a self-evolving system. We have clarified this in the revised manuscript to avoid possible misunderstanding (page 8 lines 162–164).

Comment 9: Lines 228-229 "With our proposed method, the model can self-adjust and update the training annotations to correct the possible observer biases and generate more satisfactory segmentation results". This, too, needs a detailed explanation but ends with an argument. Here, it is necessary to explain and show how the ground truth is defined clearly.

Response: Thanks. We have added a detailed explanation on this aspect. The ground-truth labels are provided by the challenges for the CHAOS and prostate segmentation datasets. There are no discrete (hard) ground-truth labels for the QUBIQ datasets. As defined by the challenge, performance characterization of models is conducted by comparing the predictions (continuous values in [0, 1]) to the continuous ground truth labels (obtained by averaging multiple experts' annotations) through thresholding the continuous labels at different probability levels (0.1, 0.2, ... 0.8, 0.9). Then, Dice similarity coefficients (DSCs) for all thresholds are averaged to get the final metrics. For our collected clinical data, the ground-truth labels are provided by three experienced radiologists with more than 10 years of experience in breast MR image interpretation. Two radiologists delineate the breast tumor regions independently, and a third radiologist (the most experienced one) checks the two sets of annotations and makes the final decision. Information about the ground-truth label generation has been added to the "Methods" section in the revised manuscript (page 19 lines 402–406, pages 19 and 20 lines 414 and 415, page 20 lines 421 and 422, page 21 lines 444–448, and pages 21 and 22 lines 459–462).

The details of how the model self-adjusts and updates the training annotations are given in the "Methods" section (pages 22 and 23 lines 475–501). Briefly, two steps are included – a local label-filtering step and a global label-correction step. The local label-filtering step aims to find data with large label noise and calculate a consistency loss on these data to decrease the influence of the noisy labels on model training. The global label-correction step updates the highly noisy labels with the model generated pseudo-labels. We have clarified in the revised manuscript that "the model can self-adjust and update the training labels via the proposed local label-filtering and global label-correction steps to correct possible observer biases" (page 13 lines 273–275). Finally, we want to clarify that the ground-truth labels are not utilized during the updating of training annotations. In other words, for model training, we have no access to high-quality ground-truth labels for data containing noisy labels, and only the noisy labels are utilized.

Comment 10: Lines 257-260 "GGH", "GPPH", and "HPPH" should be spelled out here.

Response: Thanks. In the revised manuscript, we have added the corresponding full spellings when the three datasets are first mentioned (page 13 lines 279–281). "GGH" refers to Guangdong General Hospital. "GPPH" is Guizhou Provincial People's Hospital. And "HPPH" is Henan Provincial People's Hospital.

Comment 11: I am very curious about how line 301 "orderly and selective manner" is, and it should be explained in detail and clearly.

Response: Sorry for the confusion. The orderly and selective manner of label updating is mentioned here to show the improvements of AIDE when compared to the pseudo-label method, which updates all the unlabeled data simultaneously during network learning. At the end of each training epoch, we calculate the Dice similarity coefficients (DSCs) between the network predictions in the current epoch and the pseudo-labels generated in the previous epochs for the noisily labeled data. We rank the samples according to the calculated DSC values and select the 25% of samples that have the lowest values for label updating every 10 epochs or if the epoch number is smaller than the defined warm-up epoch number. Therefore, "orderly" is reflected in two aspects. Firstly, we update the labels according to the rank of DSC values. Second, we update labels only in defined epochs. "Selective" means we only update a certain percentage of noisy labels instead of all. We have added explanations of these aspects in the revised manuscript (pages 16 lines 350–354).

Comment 12: Line 306 "in an effective and progressive manner," has no scientific explanation or logical interpretation in the paper.

Response: To address your concern, we have provided more details (pages 16 and 17 lines 359–364). Here, the effectiveness of the proposed pseudo-label generation method is validated by the good performance our method achieves on different tasks. We call it a progressive pseudo-label generation manner for two reasons. First, as shown in Algorithm 1, there is a weight hyper-parameter in the loss equations (λ_q in L1 and L2) that controls the contributions of the segmentation loss and consistency loss to the final loss for network parameter updating. This weight hyper-parameter changes with the epoch number from 0 to the maximum value of 1. As a result, the contribution of the consistency loss increases progressively, and this consistency loss is related to the local pseudo-labels generated from the augmented inputs. Second, the progressive manner can be understood from our designed two-step strategy. The local label-filtering step in each iteration only utilizes the temporal pseudo-labels to calculate the loss without actually modifying the labels, whereas the global label-correction step in each epoch (when the updating criterion is met) directly corrects the selected noisy labels. We clarified our statement in the revised manuscript to avoid misunderstanding (pages 16 and 17 lines 359–364).

Comment 13: Line 428 "Here we show that the same network memorization pattern exists for medical image segmentation." No explanation or evidence appears in the related medical image at all in this paper.

Response: We have provided more details to address your concern (pages 23 and 24 lines 502–521 in the revised manuscript and page 5 lines 161–179 in the revised supplementary information). The network memorization pattern refers to the fact that deep neural networks tend to learn simple patterns first and then gradually memorize all samples, with the real data examples being easier to fit than noise⁴⁻⁶. In our experiments, we find an overall positive correlation between the Dice similarity coefficients (DSCs) evaluating the similarity between the network outputs and noisy labels and the DSCs evaluating the similarity between the noisy labels and high-quality labels for the training set within the initial 10 epochs (Fig. R1a). In other words, within the considered

training period, the network can learn the patterns of the samples that contain low label noise, although it cannot memorize well those samples that contain large label noise. We also checked the model performance in the last 10 epochs and find that all the samples are well memorized (Fig. R1b). We need to correct the highly noisy labels before the network memorizes them well. Accordingly, we designed a specific noisy label updating schedule – we update the suspected noisy labels if the training epoch number is smaller than the defined warm-up epoch number and every 10 epochs thereafter. The first criterion is raised according to the above-mentioned network memorization behavior we observe. The second criterion is raised because, after certain epochs, the performances of the networks become relatively stable, and there is no need to update the labels so frequently. Therefore, in our experiments, we update the labels every 10 training epochs. We have clarified these descriptions in the revised manuscript (pages 23 and 24 lines 502–521).

Fig. R1 | Relationships between model memorization capability (DSCs calculated between network outputs and noisy labels (vertical axis)) and noisy label accuracy (DSCs calculated between noisy labels and high-quality labels (horizontal axis)) for the training set of the CHAOS dataset (training with 30 samples containing high-quality labels and 301 samples containing low-quality noisy labels utilizing the conventional fully-supervised learning method) in the first 10 epochs (a) and in the last 10 epochs (b). Dots represent the DSCs calculated and dashed lines indicate the linear regression results.

Responses to Reviewer #2

Comment 1: Overall, the paper is well written, code is available and I think it makes an interesting effort trying to unify three different but related problems, under the same umbrella. Moreover, the experimental results in a variety of datasets suggest that the proposed framework actually alleviates the problem of limited annotations in several scenarios. I believe a work like this would be of interest for the machine learning, computer vision and medical imaging communities.

Response: Many thanks for the positive comments and encouragements. All the comments have been carefully addressed and the corresponding revisions have been made to address your concerns. We greatly appreciate the constructive advices, which have helped us further improve our manuscript.

Comment 2: Novelty – Even though I like the effort made by the authors to unify the 3 problems (SSL, UDA and NLL) under the same umbrella, I wonder if the incremental methodological novelty with respect to existing co-training and pseudo-label methods is enough for a journal like Nature Communications.

Response: We apologize that we did not highlight enough the essential methodological differences between our method and the existing co-training and pseudo-label methods. As kindly noticed by the reviewer, this work is trying to address the challenging label-reliance issue in a robust and efficient way by unifying three different but related problems under the same umbrella. Furthermore, another major novelty of our work is the orderly and selective label updating framework designed according to an observed small-loss phenomenon or network memorization behavior, which is critical for the model performance. To the best of our knowledge, this has never been considered in the existing methods. Particularly, at the end of each training epoch, we calculate the Dice similarity coefficients (DSCs) evaluating the similarity between the network predictions in the current epoch and the pseudo-labels generated in the previous epochs for the noisily labeled data. We update a certain percentage of the noisy labels according to the rank of the DSC values in some pre-defined epochs. The existing co-training (co-teaching) methods do not consider pseudo-labels, whereas the existing pseudo-label methods do not have updating strategies that are designed specifically to noisy labels. Both can lead to deteriorated model performance when large quantities of noisy labels exist, especially for image segmentation tasks when labels are difficult to obtain. We have polished our paper to emphasize these essential and different contributions and hope our revision could address your concerns (pages 4 and 5 lines 79–91, and pages 16 and 17 lines 349–364).

Comment 3: Missing "theoretical" evaluations – In line 87 the authors mentioned that "theoretical analysis is performed to evaluate the effectiveness of AIDE...." However, I do not see any theoretical analysis, proof or demonstration supporting the claims made by the authors. On the contrary, most of the validation seems to be experimental. Am I missing something here?

Response: Very sorry for the confusion. Please kindly refer to section “Enhancement of AIDE compared to conventional fully-supervised learning”, in which we provide an analysis as to why our method can solve the issues we are encountering from three perspectives (pages 7 and 8 lines

145–170). Firstly, the local label-filtering step in each iteration forces the networks to concentrate more on the image content and circumvent the negative effects of highly noisy labels. Secondly, the global label-correction step empowers the networks with a self-evolving capability by finding useful information from their own predictions. Lastly, the cross-modal co-optimization learning strategy prevents error propagation and accumulation in one network and reduces the risk of network over-fitting to its own predicted pseudo-labels. We conclude that using the terminology “theoretical” to refer to these methodological justifications/explanations may not be fully appropriate. Therefore, we have removed the term “theoretical” to avoid any confusion in the revised manuscript.

Comment 4: SOTA discussion – Since most of the public datasets used by the authors are associated with challenges, I would have expected a discussion positioning the accuracy obtained by AIDE with respect to the state of the art in every problem, not in the tables but at least in the Discussion section. How far are the results obtained from the winners of such competitions for example? Or from more modern methods which used the same datasets?

Response: Many thanks for your constructive advice. As suggested, additional discussions have been added (page 10 lines 214–219, page 12 lines 248–251, page 12 lines 264–267, and page 15 lines 315–319).

It is worth to be noted that the top-ranking methods for these challenges are all fully-supervised methods with considerable high-quality labeled data, while ours are trying to handle the three issues related to imperfect training dataset (SSL, UDA, and NLL). So, inherently, it is not fair to directly compare our method to them. But it is interesting to check if our method can achieve comparable performance with only partial or even no high-quality labels to these fully-supervised methods.

For the semi-supervised task of liver segmentation on the CHAOS dataset, we submit our results generated by AIDE that is trained with 10 training cases (1 labeled case and 9 unlabeled cases) for evaluation online, and achieve an average DSC of 0.831 on the test data. The fully-supervised methods trained on 20 labeled cases achieved a DSC range of [0.63, 0.95] (as reported in the summary paper of the CHAOS challenge⁷). It can be seen that our method trained with only 1 high quality label and 9 unlabeled cases could achieve a reasonably good result compared to these trained with 20 high-quality labeled cases.

For the unsupervised-domain adaptation task of prostate segmentation, NCI-ISBI 2013 does not accept submissions anymore and PROMISE12 has not provided us the feedback after more than 2 months. Therefore, alternatively, we choose to compare our results on PROMISE12 with the published measurement indicators. As the ground-truth labels of the test set are not publicly available, we can only report the results on our randomly selected validation data. Nevertheless, our results (DSCs larger than 0.74) fit well in the range of the reported values [0.71, 0.90]⁸. Since no target domain training labels (no labels of the PROMISE12 data) are utilized, the performance of our method is acceptable compared to these fully-supervised ones.

For the noisy label learning task on QUBIQ datasets, the challenge is still ongoing, and it accepts the results of the validation set for the time being. After submitting our results generated by AIDE trained with only one set of annotations from one annotator for each task, we obtain a DSC of 0.932 on prostate segmentation, a DSC of 0.866 on brain growth segmentation, a DSC of 0.937 on brain tumor segmentation, and a DSC of 0.898 on kidney segmentation. These results are

comparable to those achieved by utilizing multiple sets of annotations provided by different annotators.

For breast tumor segmentation, Zhang et al. built a hierarchical convolutional neural network framework that contains three fully convolutional neural networks to segment breast tumors from DCE-MR images, and they achieved a mean DSC value of 72%¹. Qiao et al. reported a mean DSC value of 78% with 3D U-Net when single-phase DCE-MR images were utilized². Both models were trained with hundreds of high-quality labeled training cases. Our results (73.1% on the HPPH dataset achieved by AIDE trained with 25 labeled and 247 unlabeled cases) are comparable to these reported by state-of-the-art fully-supervised methods.

In summary, our method achieves promising performances on the investigated datasets when compared to the respective fully-supervised baseline methods. It should be noted that for the CHAOS and QUBIQ datasets, although we have submitted our results to the online servers for evaluation, we cannot get the ranking results as the rankings are conducted by averaging the results on different tasks instead of concentrating on the relevant segmentation tasks. In the revised manuscript, we have added the content discussing the performance of our method when compared to that achieved by existing methods (page 10 lines 214–219, page 12 lines 248–251, page 12 lines 264–267, and page 15 lines 315–319).

Comment 5: Pseudo-label baseline – In table 1, it is not clear to me what is the difference between the pseudo-label baseline and the settings like S10, where AIDE is "No" and you use LQ annotations. As far as I understand, LQAs are generated as pseudo-labels using the HQ labels for training. Thus, I do not see the difference with a vanilla pseudo-label approach. I may be missing something here because numerical results are different, but please clarify this in the manuscript.

Response: Sorry for the confusion. For our settings like S06 and S10 in Table 1, the pseudo-labels are never updated. These pseudo-labels are considered as true labels during the whole network training process. For the pseudo-label baseline, according to the paper⁹, the pseudo-labels are updated during network training. We have clarified this in the revised manuscript (page 9 lines 195–197).

Comment 6: Why is the pseudo-label baseline only used when comparing with SSL, but not as a UDA baseline in Figure 3? Couldn't it be applied in that setting?

Response: Thanks for the suggestion. Yes, the pseudo-label method can be used as a UDA baseline. Table R3 lists the results. It can be observed that the performance is improved when compared to the conventional fully-supervised learning approach with pseudo-labels of the target domain data generated by the pretrained models. Overall, our method achieves better results than these two methods. We have modified the manuscript to include part of this information (page 12 lines 249–251) and listed the results in the revised supplementary file (page 4 lines 141–147 in the supplementary information and Supplementary Table 4).

Table R3 | Segmentation results of comparison methods for the unsupervised domain adaptation task

Source domain	Target domain	Pseudo-label ⁹	Co-teaching ¹⁰	DSC (%)	RAVD (%)	ASSD (mm)	MSSD (mm)
Domain 1	Domain 2	Y	N	53.9	41.5	4.39	17.1
Domain 1	Domain 3	Y	N	60.5	45.5	4.16	14.9
Domain 2	Domain 1	Y	N	70.5	32.4	3.33	14.8
Domain 2	Domain 3	Y	N	58.0	38.1	6.38	20.2
Domain 3	Domain 1	Y	N	82.3	18.0	2.02	10.2
Domain 3	Domain 2	Y	N	78.6	31.5	3.25	21.3
Domain 1	Domain 2	N	Y	58.1	49.2	4.28	17.0
Domain 1	Domain 3	N	Y	62.2	42.4	3.87	14.8
Domain 2	Domain 1	N	Y	74.1	26.9	2.87	14.2
Domain 2	Domain 3	N	Y	66.7	43.9	3.57	14.6
Domain 3	Domain 1	N	Y	81.9	8.88	2.43	15.8
Domain 3	Domain 2	N	Y	88.1	11.1	1.33	7.51

Comment 7: Does the algorithm enforce in any way an explicit differentiation in case of UDA and SSL scenarios between the pseudo-labels and the high quality ones? As far as I understand, this is not explicitly enforced because accurately annotated samples are selected in lines 15 and 16 of the Algo1, by looking at the segmentation loss. If this is not explicitly enforced, aren't you losing the possibility of taking advantage of those labels that you know are of high quality? I may be missing something here. Please clarify this point.

Response: Thanks for raising this inspiring question. Yes, we do differentiate between the pseudo-labels and the high-quality labels for SSL and UDA. For these two tasks, the high-quality labels are always kept unchanged during network training. When a high-quality label is selected for label updating by the model, we just keep it and update the remaining selected pseudo-labels, i.e. we do not further select another one to replace the one with a high-quality label. We have clarified this information in Algorithm 1 lines 25 and 26.

Comment 8: Also related to lines 15 and 16, it is not clear to me what are the arguments of the argmin. In particular, what is $d \in \{d' : |d'| < 0.5B\}$?

Response: Sorry for the confusion. Lines 15 and 16 in Algorithm 1 (lines 14 and 15 in the revised algorithm) target to find the data samples with smaller label noise in the training sample batch. The expression $d \in \{d' : |d'| < 0.5B\}$ means we select half of the samples in the batch with smaller segmentation losses. $|d'|$ is the number count of the dataset d' and B is the batch size. This is similar to lines 22 and 23, which try to find the data cases instead of samples with larger label noise (smaller Dice similarity coefficients) in the whole training set. We have clarified these points in the algorithm (lines 14, 15, 22, and 23 in the revised algorithm).

Comment 9: In line 411, L_{seg} is defined as a metric which takes 2 segmentation masks as arguments. However, when used in Algo1 (lines 15, 16, 17, 18), it receives as a parameter a batch d' and the weights w_1 or w_2 . Please clarify if you are overloading the operator and, in that case,

Algorithm 1 | Cross-Model Self-Correcting Mechanism of AIDE.

```
1: Input Net1 (N1 with parameter  $w_1$ ) and Net2 (N2 with parameter  $w_2$ ), learning rate lr, training
   set D, batch size B, number of augmentations K, Sharpening temperature T, Segmentation
   loss function  $L_{\text{seg}}$ , Consistency loss function  $L_{\text{con}}$ , Loss weight  $\lambda$ , Label update ratio R, Label
   update criterion U, Warm-up epoch  $q_w$ , and total epoch Q
2: for  $q = 1, 2, \dots, Q$  do
3:   Shuffle training set D into  $|D|/B$  min batches;
4:   for  $n = 1$  to  $|D|/B$  do
5:     Fetch batch inputs  $d = ((x_b, y_{b1}, y_{b2}); b \in (1, \dots, B))$  from D
      ( $y_{b1}, y_{b2}$ : references for N1 and N2, initialized to be the low-quality annotations);
6:     for  $k = 1$  to K do
7:       Obtain  $\hat{x}_{b,k} = \text{Augment}(x_b)$ 
8:     end for
9:      $\bar{y}_{b1} = \frac{1}{K} \sum_k (N_1(y | \hat{x}_{b,k}; w_1))$ 
10:     $\hat{y}_{b1} = \text{Sharpen}(\bar{y}_{b1}, T)$ 
11:     $(\hat{y}_{b1,m} = \frac{\exp(\frac{\bar{y}_{b1,m}}{T})}{\sum_n \exp(\frac{\bar{y}_{b1,n}}{T})}$  with  $m$  and  $n$  referring to the categories)
12:     $\bar{y}_{b2} = \frac{1}{K} \sum_k (N_2(y | \hat{x}_{b,k}; w_2))$ 
13:     $\hat{y}_{b2} = \text{Sharpen}(\bar{y}_{b2}, T)$ 
14:    Obtain  $d_{s1} = \text{argmin}_{d' \in d: |d'| \leq 0.5B} L_{\text{seg}}(d'; w_1)$  //select 50% small loss samples in the batch
15:    Obtain  $d_{s2} = \text{argmin}_{d' \in d: |d'| \leq 0.5B} L_{\text{seg}}(d'; w_2)$  //select 50% small loss samples in the batch
16:    Obtain  $L1 = L_{\text{seg}}(N_1(y | d_{s2}; w_1), y_{d_{s2}}) + (1.0 - \lambda_q) L_{\text{seg}}(N_1(y | d_{l2}; w_1), y_{d_{l2}}) +$ 
       $\lambda_q L_{\text{con}}(N_1(y | d_{l2}; w_1), \hat{y}_{d_{l2}})$ ,  $d_{l1} = d / d_{s1}$ 
17:    Obtain  $L2 = L_{\text{seg}}(N_2(y | d_{s1}; w_2), y_{d_{s1}}) + (1.0 - \lambda_q) L_{\text{seg}}(N_2(y | d_{l1}; w_2), y_{d_{l1}}) +$ 
       $\lambda_q L_{\text{con}}(N_2(y | d_{l1}; w_2), \hat{y}_{d_{l1}})$ ,  $d_{l2} = d / d_{s2}$ 
18:    Update  $w_1 = w_1 - \text{lr} \nabla L1$ 
19:    Update  $w_2 = w_2 - \text{lr} \nabla L2$ 
20:  end for
21:  Update  $\lambda_q = \min\{\lambda * (q / q_w)^2, \lambda\}$ ,  $\lambda = 1.0$ 
22:  Obtain  $D_{l1} = \text{argmin}_{D' \in D: |D'| \leq R|D|} \text{Dice}(D'; w_1)$  //select R% small loss cases in the whole
   training set
23:  Obtain  $D_{l2} = \text{argmin}_{D' \in D: |D'| \leq R|D|} \text{Dice}(D'; w_2)$  //select R% small loss cases in the whole
   training set
24:  if U do
25:    Update  $y_{b2} = N_1(y | D_{l1}; w_1)$  if  $D_{l1}$  is not high-quality labeled
26:    Update  $y_{b1} = N_2(y | D_{l2}; w_2)$  if  $D_{l2}$  is not high-quality labeled
27:    Obtain  $\text{DSC}_{\text{train}} = \text{mean}(\text{Dice}(D; w_1), \text{Dice}(D; w_2))$ 
28:  end if
29: end for
30: Output  $w_1$  and  $w_2$  with the highest  $\text{DSC}_{\text{train}}$ 
```

what does it exactly mean?

Response: Sorry for the misleading expression. We are not overloading the operator. The arguments of the segmentation loss are the two segmentation masks (the network predictions and the references). The weights w_1 or w_2 are not the direct parameters for the loss function. As stated

in the algorithm, w_1 and w_2 are the parameters of the two networks, which determine the network predictions. “ d ” refers to the samples. We have modified the writings of loss functions in Algorithm 1 lines 16 and 17 to avoid possible misunderstanding.

Comment 10: What is q_t in line 22 of Algorithm 1?

Response: Thanks for the kind reminding. It should be q_w , i.e. the defined warm-up epoch number. We have modified it in the revised algorithm (Algorithm 1 line 21).

Comment 11: Indentation of algorithm 1 does not correspond with the beginning/end of the for loops. This should be corrected to ease understanding. Sentences which are at the same nesting level should also be at the same indentation level.

Response: Thanks for the suggestion. In the revised manuscript, we have made the revisions to Algorithm 1 accordingly.

Comment 12: Figure 3 –Figure 3 is a bit confusing. First, intuitively one would expect that red color relates to better performance while blue relates to lower performance. However, figures a to d include a mix of measures, some of which the lower the better (e.g. ASSD), and other in which the higher the better (e.g. Dice), and all of them use the same color-bar scheme. I would adjust the color bars so that blue means better performance and red means worse, or vice versa, in all the cases. Otherwise it's a bit confusing. Also, please add a title to every matrix indicating what metric it is referring to. That will help the readers and simplify their work, avoiding the need to check the figure caption all the time.

Response: Sorry for the confusion and thanks for the advice. We have modified the figure as suggested such that red color always indicates better performance. Also, we have added the subtitles.

Fig. R2 | Fig. 3 | Results of prostate segmentation for UDA. **a, b, c, and d,** The four evaluation metrics, DSC (%), RAVD (%), ASSD (mm), and MSSD (mm). In each subfigure, the left and right mappings indicate that the networks are trained without and with the proposed AIDE. D1, D2, and D3 refer to Domains 1, 2, and 3. In each mapping, the vertical axis indicates the dataset utilized to train the models and the horizontal axis indicates the dataset utilized to test the models. **e,** Example segmentation results when transferring models between domains. GT stands for ground truth, referring to the high-quality annotations. ‘Conven’ is conventional, indicating that the results are generated by a fully-supervised optimization method utilizing the combined dataset (high-quality labels for the source domain training data and model-generated low-quality labels for the target domain training data).

Comment 13: Rationale of the proposed label filtering method –In line 423, the authors comment that "The rationale for our filtering and correction steps is that we observed when training data containing noisy annotations, there is a positive correlation between the DSCs calculated comparing network outputs to current noisy labels and the DSCs calculated comparing current noisy labels to the high-quality labels for the training set" and they refer to Figure S11 in the Sup Mat. I do not clearly see how this relates to the rationale of the proposed method. I would like the authors to discuss it more explicitly.

Response: Following your advice, detailed discussions have been added to clarify this issue (pages 23 and 24 lines 502–521 in the revised manuscript and page 5 lines 161–179 in the revised supplementary information). The design of our label filtering and correction schedule is related to the network memorization behavior, which refers to the fact that deep neural networks tend to learn simple patterns first and then gradually memorize all the samples, with the real data examples being easier to fit than noise⁴⁻⁶. In our experiments, we find an overall positive correlation between the Dice similarity coefficients (DSCs) evaluating the similarity between the network outputs and noisy labels and the DSCs evaluating the similarity between the noisy labels and high-quality labels for the training set within the initial 10 epochs (Fig. R3a). In other words, within the considered training period, the network can learn the patterns of the samples that contain low label noise, although it cannot memorize well those samples containing large label noise. We also checked the model performance in the last 10 epochs, and found that all the samples are well memorized (Fig. R3b). We need to correct the highly noisy labels before the network memorizes them well. Accordingly, we designed a specific noisy label updating schedule – we update the suspected noisy labels if the training epoch number is smaller than the defined warm-up epoch number and every 10 epochs thereafter. The first criterion is raised according to the above-mentioned network memorization behavior we observe. The second criterion is raised because, after certain training epochs, the performances of the networks become relatively stable, and there is no need to update the labels so frequently. Therefore, in our experiments, we update the labels every 10 training epochs.

Fig. R3 | Relationships between model memorization capability (DSCs evaluating the similarity between the network outputs and noisy labels (vertical axis)) and noisy label accuracy (DSCs evaluating the similarity between the noisy labels and high-quality labels (horizontal axis)) for the training set of the CHAOS dataset (training with 30 samples containing high-quality labels and 301 samples containing low-quality noisy labels utilizing the conventional fully-supervised learning method) in the first 10 epochs (a) and the last 10 epochs (b). Dots represent the DSCs calculated and dashed lines indicate the linear regression results.

Responses to Reviewer #3

Comment 1: The authors describe AIDE, a method for training deep segmentation models on medical images. Their method is similar to co-training, optimizing two models simultaneously and using them to generate pseudo-labels. This technique appears to be novel, and based on the results presented, training using AIDE appears to improve network performance for semi-supervised tasks, and data with noisy labels. The MR images used for segmentation in this study are all 2D, which is ok in the context of the paper. The claim that medical datasets lack annotation, and that annotation is costly is also well validated in the literature. Overall, this paper presents a novel framework for reducing the annotation burden for medical image segmentation via deep learning.

Response: Many thanks to you for recognizing the significance of the work and the efforts made to address the important issue of deep learning-based medical image segmentation. We have carefully studied all your constructive comments and advices, according to which further improvements have been made to our manuscript. We hope this revision could address all your concerns.

Comment 2: The relationship between transforming unsupervised domain adaptation (UDA) and semi supervised learning (SSL) into a noisy label learning (NLL) task is described in the methods section. This relationship is essential to the readability of the paper and should have been presented early in the introduction. For example, this relationship is not clear in figure 1.

Response: Thanks for the suggestion. In the revised manuscript, we have described the relationship in the introduction for better readability (page 4 lines 82–84).

Comment 3: Overall the descriptions of the figures require further work. For example, in figure 2 the difference between the panels with the white and black background are not described.

Response: To address your concern, we have carefully modified the descriptions of all the figures and hope they are now better presented. In Figure 2, the plots with the white background are the 3D rendering of the segmentation results. The plots with the black background are the 2D segmentation results of one slice. The 3D and 2D results in each column are generated from the same experiment. The four sub-figures (a, b, c, and d) indicate four different cases.

Comment 4: The S## (example S01) and Anno# descriptions of the experiments / training setups used in the work are very confusing to the reader. This notation could be improved to give some context to the experimental setup used without referencing table 1. Maybe the AIDE vs non AIDE training results could be color coded for ease of interpretability.

Response: Many thanks for the suggestion. In the revised manuscript, we have tried to include the relevant context in the notations (Table 1). For example, we have changed “S01” to “S01_30_0_F_NP” with “30” referring to 30 training samples with high-quality labels, “0” meaning no unlabeled data were utilized, “F” indicating training with the conventional fully-supervised learning approach, and “NP” meaning no post-processing. The descriptions of “Anno#” are actually referring to the noisy labels provided by the different annotators. We have clarified this in the revised manuscript (Table 3). Furthermore, as suggested, different background colors have been utilized to indicate the results from experiments with and without AIDE (Table 1

and Table 3).

Comment 5: The figures references in the text seem to be an afterthought. The readability could be improved by referencing the specific figure sub panel more frequently (in the correct context).

Response: Thanks for the advice. We have carefully modified the manuscript. References to specific figure sub-panels have been included to improve the readability of the manuscript.

Comment 6: Descriptions of how the ground truth was generated are lacking in the text. The authors compare the annotations of multiple radiologists, but do not describe how the ground truth was derived for the high-quality labeled training set (for example was one radiologist’s annotation chosen?).

Response: Sorry for missing this important information. We have provided the details for all the datasets. Specifically, the ground-truth labels are provided by the challenges for the CHAOS and prostate segmentation datasets by summarizing the opinions of multiple annotators. There are no hard (discrete) ground-truth labels for the QUBIQ datasets. As defined by the challenge, performance characterization of the models is conducted by comparing the predictions (continuous values in $[0, 1]$) to the continuous ground-truth labels (obtained by averaging multiple experts’ annotations) through thresholding the continuous labels at different probability levels (0.1, 0.2, ... 0.8, 0.9). Then, the Dice similarity coefficients (DSCs) for all thresholds are averaged to get the final metrics. For our collected clinical data, the ground-truth labels are provided by three experienced radiologists with more than 10 years of experience in breast MR image interpretation. Two radiologists delineate the breast tumor regions independently and a third radiologist (the most experienced one) checks the two sets of annotations and makes the final decision. We have provided all this information in the revised manuscript (page 19 lines 402–406, pages 19 and 20 lines 414 and 415, page 20 lines 421 and 422, page 21 lines 444–448, and pages 21 and 22 lines 459–462).

Comment 7: The authors use a self-created architecture for segmentation. The training done in the paper is done from scratch. It is unclear how utilizing a previously trained more popular architecture (for example DeepLab) that has been pre-training on a natural image dataset may affect segmentation generalizability. Many works have shown that the use of pre-trained architectures for transfer learning produces very good results for medical image segmentation etc. It would be interesting to compare transfer learning to AIDE for segmentation.

Response: Many thanks for this inspiring comment. To address your concern, more results achieved by relevant published methods (pseudo-label⁹ and co-teaching¹⁰) have been added for comparison (Table 1 in the revised manuscript and Supplementary Table 4 in the revised supplementary information). For transfer learning, there are indeed works showing that transferring networks pretrained on natural image dataset to medical image segmentation is helpful¹¹. In the meantime, it should be noted that a finetune dataset with high-quality labels is needed for transfer learning. In our manuscript, three challenging tasks (semi-supervised learning, unsupervised domain adaptation, and noisy label learning) are unified under the same umbrella. Transfer learning might work for semi-supervised learning. For unsupervised domain adaptation and noisy label learning, it is difficult to utilize transfer learning as it is due to the lack of

accessible finetune datasets. In addition, transfer learning cannot exploit the unlabeled data or the noisily labeled data, whereas our proposed AIDE makes full use of these data. Therefore, it might not be appropriate, nor fair, to compare transfer learning to AIDE. The pseudo-label⁹ and co-teaching¹⁰ methods are more suitable for comparison in this situation, and we have reported their corresponding results (Table 1 and Supplementary Table 4).

Comment 8: To the same effect, the authors describe the AIDE technique as independent to the network architecture employed. Why was a custom architecture created instead of one already proven for segmentation?

Response: Sorry for this misleading presentation. We did not design special network architectures. For medical image segmentation, U-Net is a prevalent deep learning model employed in many existing studies^{7,12,13}. As the focus of the current manuscript is to reduce the reliance of deep learning models on labeled training data, we adopt U-Net as our baseline network architecture on purpose to validate that the proposed method is effective even for the most basic deep learning model. When multi-modal inputs are available, the overall model is still a U-Net but we utilize multiple encoders to extract features from multi-modalities, which is inspired by FuseNet¹⁴. We always utilize the same network architecture for our AIDE and the methods we compared with, for the sake of fair comparisons. We have added clarifications regarding the network architecture design in the revised manuscript (page 24 lines 523 and 524).

Comment 9: The descriptions of the experimental protocols for training are not well described. The reader has to dig deeply into the results section to understand how training was performed. For example, with the UDA results in table 2, it is unclear if the testing dataset was used for training with pseudo-labels generated via AIDE.

Response: Sorry for the confusion. We must clarify that testing data were never utilized for training, neither for our method AIDE nor for the methods we compared with. For UDA, the target domain contains separated training data and testing data even though there are no labels provided for the training data. Pseudo-labels are generated for the training data only to better train the model. We have modified the “Methods” section to include some of the necessary training details (page 19 lines 406–410, page 20 lines 429–433, and page 21 lines 443 and 444). In the meanwhile, the implementation details can be found in the supplementary information (pages 1 and 2 lines 16–71 in revised supplementary information).

Comment 10: The AIDE technique appears to be applicable to domains of medical image analysis beyond segmentation. This would enhance the appeal of the work to a greater audience.

Response: We greatly appreciate the reviewer’s suggestion as to a wider impact of the work. Investigating the applicability of our framework for other medical imaging tasks is very appealing. However, huge efforts and time are needed to accumulate enough carefully curated and widely accepted datasets for the validation of AIDE on other tasks, which may not be plausible at the current stage. Therefore, we may investigate this aspect in the future. In the third paragraph of the “Discussion” section, we have stated that we will investigate the applicability of our framework for other medical imaging tasks in the future work (page 17 lines 379–382).

Comment 11: Overall, this paper presents a novel framework for reducing the annotation burden for medical image segmentation via deep learning. In the reviewer's opinion, the readability of the work is not up to the standards of Nature Communications. The figures are not well incorporated into the write up, and the experimental setups are not apparent without close inspection and stipulation. The reviewer acknowledges that this is a complex topic, which may be hard to describe, but a hallmark of the Nature style is the readability by a greater audience.

Response: Very sorry for the encountered issues when reading the manuscript. All of our coauthors including the one from Canada (I.B.A), who is a native English speaker, have carefully proofread the revised manuscript. Furthermore, we employed professional editors from "Springer Nature Author Services" to further polish the whole manuscript. Hope the current presentation is satisfying.

References

1. Zhang, J., Saha, A., Zhu, Z. & Mazurowski, M. A. Hierarchical Convolutional Neural Networks for Segmentation of Breast Tumors in MRI With Application to Radiogenomics. *IEEE Trans. Med. Imaging* **38**, 435–447 (2019).
2. Qiao, M. *et al.* Three-dimensional breast tumor segmentation on DCE-MRI with a multilabel attention-guided joint-phase-learning network. *Comput. Med. Imaging Graph.* **90**, 101909 (2021).
3. Towards trustable machine learning. *Nat. Biomed. Eng.* **2**, 709–710 (2018).
4. Arplt, D. *et al.* A closer look at memorization in deep networks. in *International Conference on Machine Learning (ICML)* (2017).
5. Jiang, L., Zhou, Z., Leung, T., Li, L.-J. & Li, F.-F. MentorNet: Learning data-driven curriculum for very deep neural networks on corrupted labels. in *International Conference on Machine Learning (ICML)* (2018).
6. Chen, P., Liao, B., Chen, G. & Zhang, S. Understanding and utilizing deep neural networks trained with noisy labels. in *International Conference on Machine Learning (ICML)* (2019).
7. Kavur, A. E. *et al.* CHAOS Challenge - combined (CT-MR) healthy abdominal organ segmentation. *Med. Image Anal.* **69**, (2021).
8. Litjens, G. *et al.* Evaluation of prostate segmentation algorithms for MRI: The PROMISE12 challenge. *Med. Image Anal.* **18**, 359–373 (2014).
9. Lee, D.-H. Pseudo-label: The simple and efficient semi-supervised learning method for deep neural networks. *ICML 2013 Work. Challenges Represent. Learn.* 1–6 (2013).
10. Han, B. *et al.* Co-teaching: Robust training of deep neural networks with extremely noisy labels. in *Conference on Neural Information Processing Systems (NeurIPS)* (2018).
11. Zhou, Z., Sodha, V., Pang, J., Gotway, M. B. & Liang, J. Models Genesis. *Med. Image Anal.* **67**, 101840 (2021).
12. Falk, T. *et al.* U-Net: deep learning for cell counting, detection, and morphometry. *Nat. Methods* **16**, 67–70 (2019).
13. Wang, K. *et al.* Automated CT and MRI liver segmentation and biometry using a generalized convolutional neural network. *Radiol. Artif. Intell.* **1**, 180022 (2019).
14. Hazirbas, C. & Ma, L. FuseNet: Incorporating depth into semantic segmentation via fusion-based CNN architecture. in *Asian Conference on Computer Vision (ACCV)* (2016).

Reviewers' Comments:

Reviewer #1:

Remarks to the Author:

Thanks for adding a detailed explanation and replying. As a reviewer, I am sorry that the answer to my question is not clear or there are parts that I cannot fully agree with yet.

This paper lacks a sufficiently clear answer to my question as a reviewer for the most critical part.

Much effort is acknowledged, but after presenting essential novelties and problems, the solution offers only a lesser degree of performance than developing a model with sufficient data.

Reviewer #2:

Remarks to the Author:

The authors have addressed my main concerns in the revised version of the manuscript. I therefore think this paper is can be accepted for publication.

Reviewer #3:

Remarks to the Author:

AIDE: Annotation-efficient deep learning for automatic medical image segmentation

Response to referees letter

We thank all the reviewers again for spending valuable time on evaluating our manuscript. We are glad to know both Reviewers #2 and #3 are satisfied with our revision. We also appreciate Reviewer #1 for acknowledging our efforts and recognizing we have presented essential novelties and problems. To address the last confusion from Reviewer #1, we have made further clarifications.

Response to Reviewer #1

Comment: Thanks for adding a detailed explanation and replying. As a reviewer, I am sorry that the answer to my question is not clear or there are parts that I cannot fully agree with yet. This paper lacks a sufficiently clear answer to my question as a reviewer for the most critical part. Much effort is acknowledged, but after presenting essential novelties and problems, the solution offers only a lesser degree of performance than developing a model with sufficient data.

Response: Thank you very much for acknowledging our efforts and recognizing we have presented essential novelties and problems. We feel sorry that our presentations still leave you some confusions. To clarify your confusion, we have further added discussions and highlighted our contributions. Please kindly see following the details.

Existing high-performance deep learning methods typically rely on large training datasets with high-quality manual annotations, which are difficult to obtain in many clinical applications (page 2 lines 26 and 27). To address this issue, we introduce an open-source framework AIDE to handle imperfect training datasets (page 4 lines 78 and 79). We have evaluated our proposed method on three public datasets, including datasets having limited annotations, datasets lacking target domain annotations, and datasets containing noisy annotations. Experimental results show that our proposed method AIDE surpasses conventional fully-supervised models by presenting better performance on these open datasets possessing scarce or noisy annotations. We further test AIDE in a real-life case study for breast tumor segmentation. Three datasets containing 11,852 breast images from three medical centers are employed, and AIDE, utilizing 10% training annotations, consistently produces segmentation maps comparable to those generated by fully-supervised counterparts or provided by independent radiologists.

So as kindly noticed by you, instead of building a more sophisticated model for fully-supervised learning, we develop an open-source framework that can work properly without sufficient labeled data, so as to alleviate the reliance on the time-consuming and expensive manual annotations when applying AI to medical imaging. We sincerely hope this response can address all your concerns.